# Absolute Calibration method for FMCW Cloud Radars based on corner reflectors

Felipe Toledo[1], Julien Delanoë[2], Martial Haeffelin[3], Jean-Charles Dupont[4], Susana Jorquera[2], and Christophe Le Gac[2]

[1]Laboratoire de Météorologie Dynamique, École Polytechnique, Institut Polytechnique de Paris, 91128 Palaiseau, France
[2]LATMOS/IPSL, UVSQ Université Paris-Saclay, Sorbonne Université, CNRS, Guyancourt, France
[3]Institut Pierre Simon Laplace, École Polytechnique, CNRS, Institut Polytechnique de Paris, 91128 Palaiseau, France
[4]Institut Pierre-Simon Laplace, École Polytechnique, UVSQ, Université Paris-Saclay, 91128 Palaiseau, France

**Correspondence:** Felipe Toledo (ftoledo@lmd.polytechnique.fr)

**Abstract.**

This article presents a new Cloud Radar calibration methodology using solid reference reflectors mounted on masts, developed during two field experiments held in 2018 and 2019 at the SIRTA atmospheric observatory, located in Palaiseau, France, in the framework of the ACTRIS-2 research and innovation program.

The experimental setup includes 10 cm and 20 cm triangular trihedral targets installed at the top of 10 m and 20 m masts, respectively. The 10 cm target is mounted on a pan-tilt motor at the top of the 10 m mast to precisely align its boresight with the radar beam. Sources of calibration bias and uncertainty are identified and quantified. Specifically, this work assesses the impact of receiver compression, temperature variations inside the radar, frequency dependent losses in the receiver IF, clutter and experimental setup misalignment. Setup misalignment is a source of bias previously undocumented in the literature, that can have an impact on the order of tenths of dB in calibration retrievals of W band Radars.

A detailed analysis enabled the quantification of the importance of each uncertainty source to the final cloud radar calibration uncertainty. The dominant uncertainty source comes from the uncharacterized reference target, reaching 2 dB. Additionally, the analysis revealed that our 20 m mast setup with an approximate alignment approach is preferred to the 10 m mast setup with the motor-driven alignment system. The calibration uncertainty associated with signal-to-clutter ratio of the former is ten times smaller than for the latter.

Following the proposed methodology it is possible to reduce the added contribution from all uncertainty terms, excluding the target characterization, down to 0.4 dB. Therefore, this procedure should enable to achieve calibration uncertainties under 1 dB when characterized reflectors are available.

Cloud radar calibration results are found to be repeatable when comparing results from a total of 18 independent tests. Once calibrated the cloud radar provides valid reflectivity values when sampling mid-tropospheric clouds. Thus we conclude that the method is repeatable and robust, and that the uncertainties are precisely characterized. The method can be implemented under different configurations as long as the proposed principles are respected. It could be extended to reference reflectors held by other lifting devices such as tethered balloons or unmanned aerial vehicles.

# 1   Introduction

Clouds remain to this day one of the major sources of uncertainty in future climate predictions (Boucher et al., 2013; Myhre et al., 2013; Mülmenstädt and Feingold, 2018). This arises partly from the wide range of scales involved in cloud systems, where a knowledge of cloud micro-physics, particularly cloud-aerosol interaction, is critical to predict large scale phenomena such as cloud radiative forcing or precipitation.

To address this and other related issues, the ACTRIS Aerosols, Cloud and Trace Gases Research Infrastructure is establishing

an state of the art ground based observation network (Pappalardo, 2018). Within this organization, the Centre for Cloud Remote Sensing CCRES is in charge of creating and defining calibration and quality assurance protocols for the observation of Cloud properties across the complete network.

One of the key instruments for cloud remote sensing stations is the Cloud Radar. Cloud radars enable retrievals of several relevant parameters for cloud research, including but not limited to liquid water and ice content profiles, cloud boundaries, cloud

fraction, precipitation rate and turbulence (Fox and Illingworth, 1997; Hogan et al., 2001; Wærsted et al., 2017; Dupont et al., 2018; Haynes et al., 2009). Additionally, recent studies revealed the potential of cloud radars to support a better understanding of fog processes (Dupont et al., 2012; Boers et al., 2013; Wærsted et al., 2019).

However, calibration remains a crucial factor in the reliability of radar retrieved data (Ewald et al., 2019). Systematic differences of 2 dB have already been observed, for example, between the satellite based radar CloudSat and the Lindenerg

MIRA (Protat et al., 2009). This is a very important issue, since calibration errors as small as $1\ dB$ would already introduce uncertainties in liquid water and ice content retrievals in the order of 15-20% (Fox and Illingworth, 1997; Ewald et al., 2019).

Since the objective of the CCRES is to guarantee a network of high quality observations, it is essential to develop standardized and repeatable calibration methods for its instrumental network.

This paper presents an absolute calibration method for W band radars. It has been developed based on results from two

experimental calibration campaigns performed at the SIRTA Atmospheric Observatory, located in Palaiseau, France (Haeffelin et al., 2005). The SIRTA observatory hosts part of the ACTRIS CCRES infrastructure. For the experiments we used a BASTA-Mini W band Frequency Modulated Continuous Wave (FMCW) Radar, with scanning capabilities (Delanoë et al., 2016). Nevertheless, the principles, procedures and limitations presented here should be applicable for any radar with similar characteristics, even when operating in another frequency band.

The method consists on an end-to-end calibration approach, consisting in retrieving the radar calibration coefficient by sampling the power reflected from a reference reflector mounted on top of a mast (Chandrasekar et al., 2015). A detailed analysis of uncertainty and bias sources is performed, with the objective of determining how to improve the experiment to reach a calibration uncertainty of lower than $1\ dB$. This low uncertainty in the calibration would not only be useful for high quality retrievals, but also enables the use of the radar as a reliable reference for calibration transfer to other ground or space

based cloud radars (Bergada et al., 2001; Protat et al., 2011; Ewald et al., 2019).

The article is structured as follows: Section 2 present the equations and theoretical considerations involved in the calibration exercise. Section 3 shows the experimental setup, complemented by section 4 where the experimental procedure and data treat-

ment is presented. Section 5 presents an analysis of the sources of uncertainty and bias involved in our calibration experiment. Section 6 presents the final calibration results, the uncertainty budget and an analysis of the variability in the calibration bias
correction, followed by the conclusions.

## 2   Equations used in Radar Calibration

The absolute calibration of a radar consists in determining the RCS Calibration Term $C_\Gamma$ and the Radar Equivalent Reflectivity Calibration Term $C_Z$. They enable the calculation of Radar Cross Section $\Gamma(r)$ (RCS) or Radar Equivalent Reflectivity $Z_e$ respectively, from the power backscattered by a punctual or distributed target towards the radar (Bringi and Chandrasekar,
2001).

Equation (2a) presents an expression for the RCS calibration term $C_\Gamma(T, F_b)$ of a FMCW radar as a function of its internal parameters. The deduction of this expression is shown in the supplementary material. $G_t$ and $G_r$ are the maximum gain of the transmitting and receiving antennas respectively, dimensionless. $\lambda$ is the wavelength of the carrier wave in meters and $p_t$ is the power emitted by the radar in milliwatts.

The gain of solid state components changes with variations in their temperature. Thus we make this dependence explicit in the receiver loss budget $L_r(T, F_b)$ and in the transmitter loss budget $L_t(T)$. Loss budgets are the product of all losses divided by the gain terms at the end of the receiver or emitter chain, and are dimensionless.

Additionally, a range dependence is included in $L_r(T, F_b)$ to account for variations in the receiver IF loss for different beat frequency $F_b$ values. The beat frequency in FMCW radars is proportional to the distance between the instrument and
the backscatterer element (Delanoë et al., 2016). Thus, changes in the IF loss for different beat frequencies introduce a range dependent bias. For the 12.5 meter resolution mode used in this calibration exercise, $F_b$ ranges between 168 and 180 $MHz$, and can be related to $r$ (in meters) using Eq. (1).

$$r = 500 \cdot (F_b - 168[MHz]) \tag{1}$$

In theory, $C_\Gamma(T, F_b)$ can be calculated by characterizing the gains and losses of every component inside the radar system
and adding them. This can be very challenging, depending on the complexity of the radar hardware and the available radio frequency analysis equipment. In addition, with this procedure it is not possible to quantify losses due to interactions between different components, specially changes in antenna alignment or radome degradation (Anagnostou et al., 2001). This motivates the implementation of an end-to-end calibration, which consists on the characterization of the complete radar system at once by using a reference reflector and Eq. (2b).

$$C_\Gamma(T, F_b) = 10 \log_{10} \left( \frac{L_t(T) L_r(T, F_b)(4\pi)^3}{G_t G_r \lambda^2 p_t} \right) \tag{2a}$$

$$\Gamma(r) = C_\Gamma(T, F_b) + 2L_{at}(r) + 40 \log_{10}(r) + P_r(r) \tag{2b}$$

Equation (2b) links the calibration term $C_\Gamma(T, F_b)$ to the RCS $\Gamma(r)$ of a target at a distance $r$. $\Gamma(r)$ is expressed in $dBsm$ units (decibels referenced to a square meter), $L_{at}(r)$ is the atmospheric attenuation between the object and the radar in $dB$, which can be calculated using a millimeter-wave attenuation model (for ex. (Liebe, 1989)), $P_r(r)$ is the power received from the target in $dBm$ and $C_\Gamma(T, F_b)$ is the RCS calibration term in $dB(m^{-2}\ mW^{-1})$. The $dB(m^{-2}\ mW^{-1})$ unit is the abbreviation of $dB$ referenced to $1\ m^{-2}\ mW^{-1}$. The units in the RCS calibration term compensate the radar power units, guaranteeing the retrieval of physical RCS values. The explicit temperature and range dependency of the calibration term has the function of compensating gain changes in $P_r(r)$ introduced by temperature effects and variations in the IF loss with distance.

This principle can be used in an end-to-end calibration by installing a target with a known RCS $\Gamma_0$ at a known distance $r_0$ and sampling the power $P_r(r_0)$ reflected back to calculate $C_\Gamma(T, F_b)$. However, some additional considerations must be made to perform this retrieval.

In Eq. (2a) we state that the calibration value has a temperature and a range dependency. Experimental results indicate that the temperature dependency of $C_\Gamma(T, F_b)$ can be approximated by a linear relationship, as shown in Eq. (3). Here $n$ is the temperature dependency term in $dB\ °C^{-1}$, $T$ the internal radar temperature in $°C$ and $T_0$ a reference temperature value in $°C$. More details about the temperature correction can be found in Sect. (5.4).

The range dependence of $C_\Gamma(T, F_b)$ is treated independently by defining a IF loss correction function $f_{IF}(F_b)$, in $dB$. This function is introduced to compensate for relative loss variations at different IF frequencies. The IF loss correction function is studied in Sect. 5.5.

From the aforementioned observations, we divide $C_\Gamma(T, F_b)$ in three components, shown in Eq. (3). This separation consists of a constant calibration coefficient $C_\Gamma^0$, in $dB(m^{-2}\ mW^{-1})$, and the two correction functions $n(T - T_0)$ and $f_{IF}(F_b)$.

$$C_\Gamma(T, F_b) = C_\Gamma^0 + n(T - T_0) + f_{IF}(F_b) \tag{3}$$

As $f_{IF}(F_b)$ corrects for relative variations in receiver loss with distance, we define $f_{IF}(F_0) = 0$ at the IF frequency value $F_0$, associated to the reflector position $r_0$ (linked by Eq. (1)). Using this and Eqs. (2b) and (3), we get Eqs. (4a) and (4b).

$$C_\Gamma(T, F_0) = C_\Gamma^0 + n(T - T_0) \tag{4a}$$
$$C_\Gamma(T, F_0) = \Gamma_0 - 40log(r_0) - 2L_{at}(r_0) - P_r(r_0) \tag{4b}$$

Equation (4a) shows how the calibration term $C_\Gamma(T, F_0)$ at position $r_0$ is related to the calibration coefficient $C_\Gamma^0$ and the temperature correction $n(T - T_0)$. Meanwhile, Eq. (4b) indicates how experimental $P_r(r_0)$ measurements can be associated with a $C_\Gamma(T, F_0)$ value, using in-situ information to calculate $2L_{at}(r_0)$. Then, using Eq. (4a), we can compute $C_\Gamma^0$ by subtracting the temperature correction function $n(T - T_0)$. This temperature correction is derived independently in Sect. (5.4). Knowing $C_\Gamma^0$ and the temperature correction, $C_\Gamma(T, F_b)$ is calculated by adding the IF correction function, independently retrieved in Sect. 5.5.

Once $C_\Gamma(T, F_b)$ is known, we can calculate the radar Equivalent Reflectivity calibration term $C_Z(T, F_b)$, in $dB(mm^6\ m^{-5}\ mW^{-1})$, with Eq. (5a) (Yau and Rogers, 1996). This relationship assumes the radar has two identical parallel antennas with a Gaussianly shaped main lobe. $\theta$ is the antenna beamwidth in radians, $m\delta r$ is the radar distance resolution in meters and $|K| = |(\epsilon_r - 1)/(\epsilon_r + 2)|$ is the dielectric factor. This factor is related to the relative complex permittivity $\epsilon_r$ of the scattering particles, and can be calculated, for example, using the results of Meissner and Wentz (2004).

$C_Z(T, F_b)$ enables the calculation of the Radar Equivalent Reflectivity $Z_e$, in $dBZ$, of a distributed target located at a distance $r$ by using Eq. (5b). The $dBZ$ unit is usually used to express Radar Equivalent Reflectivity in logarithmic units, and is related with the linear units by $1\ dBZ = 10\ log_{10}(1\ mm^6\ m^{-3})$.

$$C_Z(T, F_b) = 10\log_{10}\left(\frac{8\ln(2)\lambda^4 10^{18}}{\theta^2\pi^6 K^2\delta r}\right) + C_\Gamma(T, r) \tag{5a}$$

$$Z_e(r) = C_Z(T, F_b) + 2L_{at}(r) + 20\log_{10}(r) + P_r(r) \tag{5b}$$

## 3 Experimental setup

Two calibration campaigns, that lasted one month each, were performed in May-June of 2018 and March-April of 2019 at the SIRTA observatory, located in Palaiseau, France (Haeffelin et al., 2005). The observatory has a 500 meter long grass field in an area free of buildings, trees or other sources of clutter, well suited to install our calibration setup, shown in Fig. 1.

The instrument used for the calibration experiments is a BASTA-Mini. BASTA-Mini is a 95 $GHz$ FMCW radar with scanning capabilities and two parallel Cassegrain antennas (Delanoë et al., 2016). The antennas are separated by 35 $cm$, and have a Fraunhofer far field distance of $\approx 50\ m$ with a Gaussianly shaped main lobe (verified experimentally in Sect. 5.2). Transmitted power is fixed to 500 $mW$, and is under constant monitoring using a diode with an uncertainty of $\approx 0.4\ dB$. The diode enable the monitoring of $L_t(T)$ variations, yet our experiments have shown that $T$ is a better indicator to capture the variability of $C_\Gamma(T, F_b)$. This is likely because internal temperature changes affect both $L_r(T, F_b)$ and $L_t(T)$ simultaneously, and therefore the information provided by the diode is not sufficient to capture the behavior of the whole system. The results of the temperature dependency study for our radar is shown in Sect. 5.4.

This radar also includes hardware to enable the tuning of the carrier wave frequency within a range of $\approx 1\ GHz$, centered at 95 $GHz$. During the experiments we fixed the BASTA-Mini base frequency at 95.64 $GHz$ to avoid any interference with the other two W band radars operating in parallel at the same site.

Our reference targets are two Triangular Trihedral Reflectors (also known as Corner Reflectors) composed by three orthogonal triangular conducting plates. Trihedral targets have a large RCS for their size and a low angular variability of RCS around their boresight (Atlas, 2002; Doerry and Brock, 2009; Chandrasekar et al., 2015). One reflector has a size parameter of 10 $cm$, with a maximum theoretical RCS at our radar operation frequency of 16.30 $dBsm$. The other is 20 $cm$ with a maximum theoretical RCS of 28.34 $dBsm$ (Brooker, 2006). These targets were mounted on top of masts B and C in Fig. 1 respectively. Only mast C was used in the 2018 campaign, while both were used in 2019.

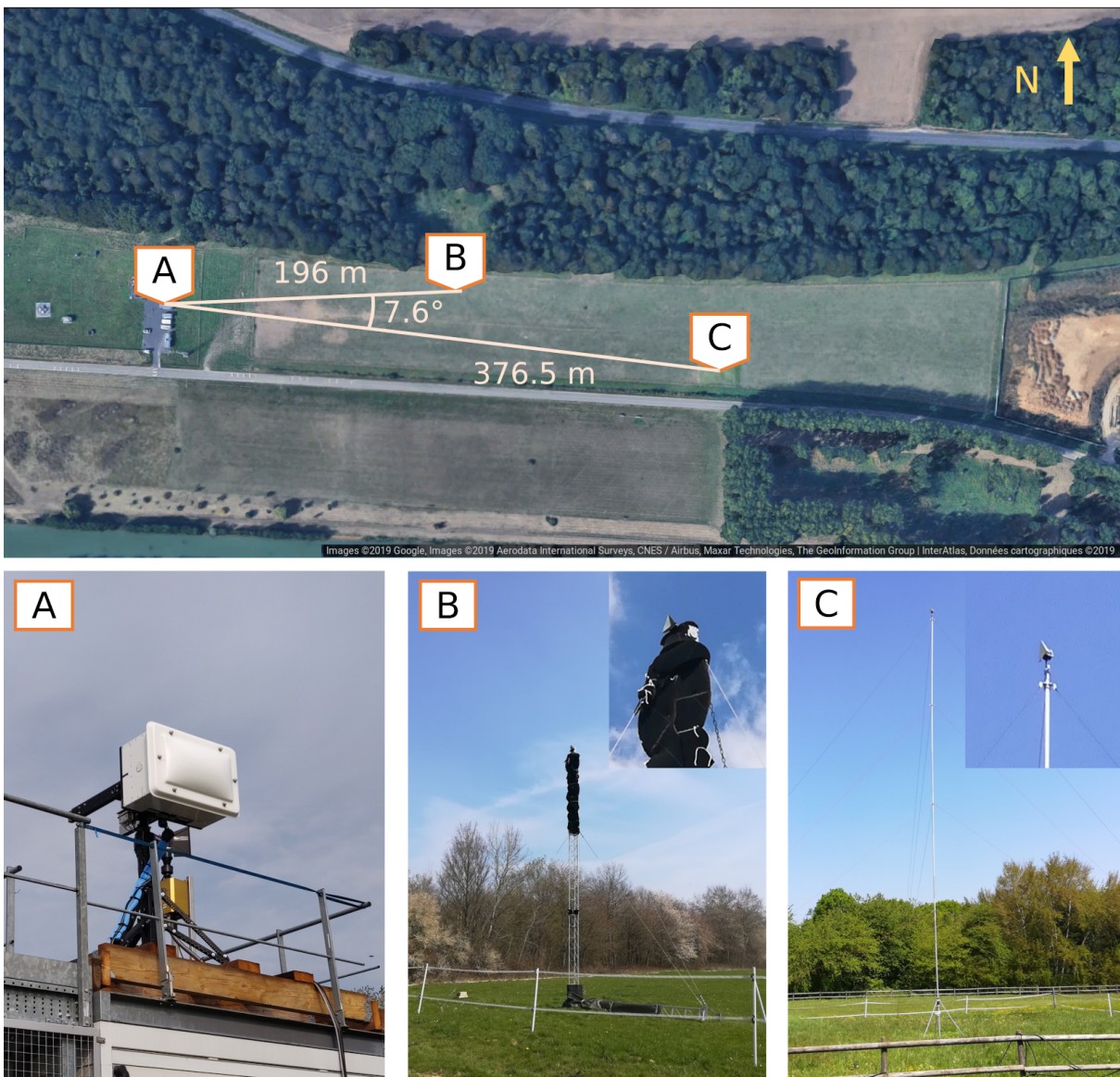

**Figure 1.** Experimental setup for 2018 and 2019 calibration experiments: (A) Scanning BASTA-Mini radar located in a reinforced platform 5 m above the ground. (B) 10 mast with a 10 cm triangular trihedral target mounted on a pan-tilt motor with an angular resolution and repeatability better than $0.1°$. This mast has microwave absorbing material wrapped to it to reduce its RCS (clutter). The 10 m mast was only installed in the 2019 calibration campaign. (C) 20 m mast with a 20 cm triangular trihedral target. The target aiming is fixed relative to the mast. This mast is used in both 2018 and 2019 calibration campaigns. Angular separation between the masts is enough to sample both targets without mutual interference.

To align the system first we aim the radar towards the approximate position of the target. Second, we aim the target by slowly changing pan-tilt angles in the motor on mast B, or axially rotating the tube of mast C to maximize the power $P_r(r_0)$ measured at the radar. Third, radar aiming is tuned around target position until the maximum reflected power is found. Finally, we repeat the second step, after which we have the system ready to sample $P_r(r_0)$.

It must be mentioned that this procedure does not guarantee a perfect alignment. In fact, it is impossible to have every element perfectly adjusted because of limits in the radar scanner resolution or uncertainties introduced when installing each element. Sections 4 and 5.6 explain how we deal with these limitations.

## 4   Methodology

This section describes the procedure followed when performing calibration experiments using the setup described in Sect. 3. The methodology has the objective of quantifying and correcting when possible all sources of uncertainty to enable a reliable estimation of the calibration terms $C_\Gamma(T, F_b)$ and $C_Z(T, F_b)$.

A challenge we found when using targets mounted on masts to estimate $C_\Gamma(T, F_b)$ is that the value of the target RCS $\Gamma_0$ may vary depending on how components are aligned. Our studies have shown that for the feasible alignment accuracy we can get when installing our setup, this effect is in the order of tenths of $dB$, and therefore not negligible. Additionally, we concluded that if we leave this uncertainty source uncorrected, we would introduce a bias in the calibration result (see Sect. 5.6).

The flow chart of Fig. 2 illustrates the calibration procedure. To quantify the bias introduced by alignment uncertainty we decided to divide each calibration experiment in $N$ iterations. Each iteration consists on a system realignment, followed by sampling of the target signal $P_r(r_0)$ for at least one hour. Then, we select the data from the contiguous hour with the lowest variability as the iteration result.

The period chosen to perform the sampling is important, because it will have an incidence on how stable is the calibration value. To minimize uncertainty it is recommended to perform calibration iterations when the atmosphere is clear, there is no rain and wind speed is under $1\ m\ s^{-1}$. However, these requirements may change depending on how robust is each setup to atmospheric conditions.

FMCW radars have a discrete distance resolution. Consequently, power measurements vs distance are resolved in finite discrete points, usually named gates. Because of this resolution limitation, power received from a point target is spread between the gates closer to its position (Doviak and Zrnić, 2006). This phenomena is known as spectral leakage. To reduce leakage BASTA-Mini uses a Hann time window (Richardson, 1978; Delanoë et al., 2016).

To correctly asses the total reflected power we set the radar resolution to 12.5 meters (chirp bandwidth of 12 MHz), and its integration time to 0.5 seconds. This resolution is high enough to accurately identify the reference reflector signal while avoiding the introduction of additional clutter from the trees located behind the mast (see Fig. (3)).

To calculate $P_r(r_0)$ we add five gates: the target gate plus two before and two after the target position. Adding more contiguous gates increase the power value by less than $0.01\ dB$, thus we conclude that these five gates concentrate almost all the power reflected back from the target.

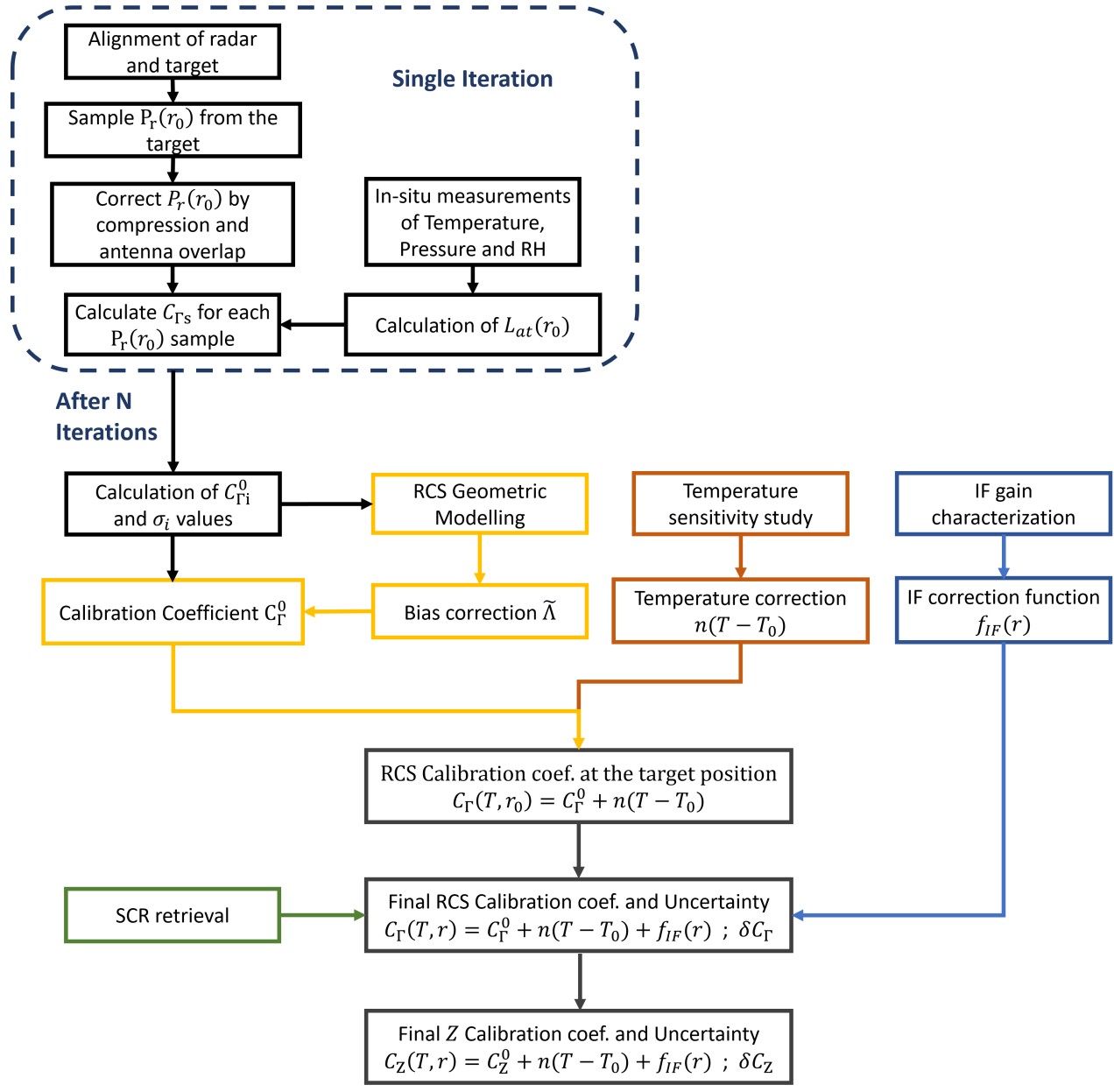

**Figure 2.** Summary of a complete calibration process. Each calibration requires repetition of system realignment and sampling steps, called iterations. During each iteration we continuously sample the power reflected from the reference target position for one hour (power corrections in Sect. 5.1). The retrieval of $N$ iterations enable the estimation of the system bias due to misalignments in the setup (Sect. 5.6). Temperature dependency is retrieved in an independent experiment (Sect. 5.4). Uncertainty introduced by clutter signals at the target location is also included in the total uncertainty budget (Sect. 5.3).

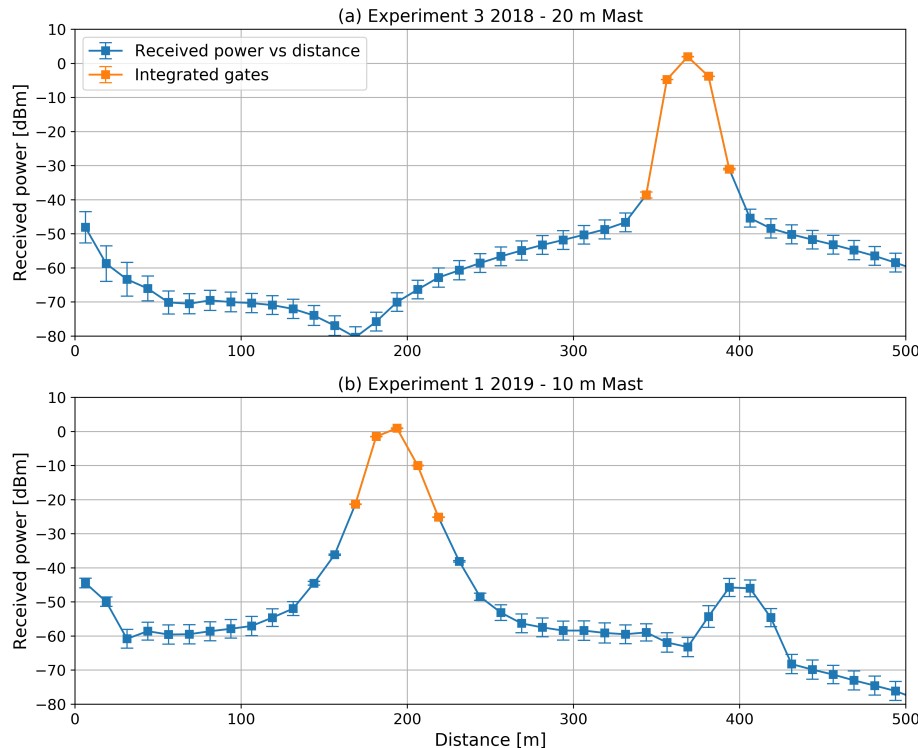

**Figure 3.** Mean profiles of received power for Experiment 5 in 2018 using the 20 m mast (a) and Experiment 1 in 2019 using the 10 m mast (b). Standard deviation at each gate is indicated with an errorbar. The gates integrated to calculate the reference reflector backscattered power $P_r(r_0)$ are marked in orange. The secondary peak of figure (b), around 400 meters, corresponds to reflections on trees behind the 10 m mast.

Then $P_r(r_0)$ is corrected considering compression effects and antenna overlap losses (Sects. 5.1 and 5.2). For each corrected $P_r(r_0)$ sample we proceed to calculate a single $C_\Gamma^0$ value with Eq. (4a) and the temperature correction function. This single sample is defined as $C_{\Gamma s}^0$ to differentiate it from the final calibration coefficient $C_\Gamma^0$ of Eq. (3). Atmospheric attenuation $L_{at}(r_0)$ is calculated using in-situ atmospheric observations and the model published by Liebe (1989).

The target effective RCS $\Gamma_0$ is calculated using a theoretical RCS model, considering the beam incidence angle on the target. Echo chamber measurements have shown that real targets RCS can be deviated from the theoretical value depending on the manufacturing precision. Our corner reflectors have an angular manufacturing precision better than $0.1°$, therefore real RCS uncertainty with respect to the model can be roughly estimated to be approximately $2\ dB$ (Garthwaite et al., 2015). Once an experimental characterization of the target becomes available, it can be used to correct any calibration bias and to reduce uncertainty by rectifying the value of $\Gamma_0$ used in the calculations.

We performed one calibration experiment with 6 iterations during the 2018 campaign using the 20 m mast. In the 2019 campaign we did two experiments: one with 10 iterations using the 10 m mast and another with 2 iterations on the 20 m mast (Fig. 1).

The retrieval of the temperature dependency coefficient $n$ and the reference temperature $T_0$ is done simultaneously with the calibration coefficient experiment, by extending the sampling period beyond one hour when using the 20 m mast. This is done to capture the temperature effect in the variability of $C^0_{\Gamma s}$, by capturing a larger part of the temperature daily cycle. The results of this experiment can be seen in Sect. 5.4. Likewise, the retrieval of the IF correction function $f_{IF}(F_b)$ is an independent experiment based on sampling noise with the radar to get the IF amplification curve of the receiver. The details of this experiment are in Sect. 5.5.

From each iteration we get a distribution of resulting $C^0_{\Gamma s}$ values with a small spread introduced by second order effects. The average value of each iteration $i$ is named $C^0_{\Gamma i}$, and its corresponding standard deviation is named $\sigma_i$. With this information we proceed to calculate the bias corrected calibration coefficient $C^0_\Gamma$, by using Eq. (6). $\tilde{\Lambda}$ is the bias correction term. The method used to calculate $\lambda$ relies on simulating the probability distribution of $\Gamma_0$ for a given set of uncertainties in the setup parameters. More detail can be found in Section 5.6 and Section S3 of the supplementary material.

$$C^0_\Gamma = \frac{1}{N} \sum_{i=1}^{N} C^0_{\Gamma i} - \tilde{\Lambda} \tag{6}$$

Equations (7a) and (7b) show the uncertainties $\delta C_\Gamma$ and $\delta C_Z$ associated with the estimation of $C_\Gamma(T, F_b)$ and $C_Z(T, F_b)$ respectively.

$\sigma_T$ is the uncertainty term associated with the temperature correction function $n(T - T_0)$.

$\sigma_{IF}$ is the uncertainty term associated with the IF loss correction function $f_{IF}(F_b)$.

The term $\sum \sigma_i^2$ comes from the averaging operation in the estimation of $C^0_{\Gamma i}$ (Eq. 6). Since the $C^0_{\Gamma i}$ terms are corrected using the temperature correction function, the uncertainty of the later must be propagated as well, hence the term $\sigma_T^2 / N$ appears.

$\sigma_\Lambda$ the uncertainty of the bias correction calculation. It is calculated from the standard deviation $\sigma_i$. This procedure is explained in Section S3 of the supplementary material.

$\sigma_{SCR}$ is the uncertainty introduced by clutter. Clutter is the presence of unwanted echoes which affect our reading of $P_r(r_0)$, coming from reflections on other objects in the environment. The method to quantify the uncertainty $\sigma_{SCR}$ uses a parameter named Signal to Clutter Ratio (SCR), explained in detail in Sect. 5.3.

$\sigma_{\Gamma_0}$ is the uncertainty of the reference target RCS. In this work we use a theoretical model to calculate the target effective RCS, which has an uncertainty of approximately 2 dB based on the manufacturing characteristics. The inclusion of an experimental characterization of the target RCS can improve the estimation of $C^0_\Gamma$ and $\delta C_\Gamma$ by reducing this uncertainty term.

$\sigma_K$ is the uncertainty in the estimation of the backscattering particles dielectric factor. Because our objective is to calculate the calibration term of the radar, we reference this value to $|K| = 0.86$, corresponding to pure water at $5\,^\circ C$ and neglect the $\delta_K$ uncertainty term. However, the value of $K$ and its uncertainty $\sigma_K$ must be considered when performing radar retrievals (e.g. Sassen (1987); Liebe et al. (1989); Gaussiat et al. (2003)).

$\sigma_A$ is the uncertainty introduced in the estimation of $\theta$ and from parallax errors and deviations from a Gaussian beam shape (Sekelsky and Clothiaux, 2002). For this work we make the assumption of parallel antennas with a Gaussian beam shape, thus we neglect this term. This problem is discussed more in depth in Section 5.2.

Since both $\sigma_K$ and $\sigma_A$ are neglected, we get $\delta C_\Gamma \approx \delta C_Z$.

$$\delta C_\Gamma(T, F_b) = \sqrt{\frac{1}{N^2}\sum_{i=1}^{N}\sigma_i^2 + \frac{\sigma_T^2}{N} + \sigma_{IF}^2 + \sigma_T^2 + \sigma_{SCR}^2 + \sigma_\Lambda^2 + \sigma_{\Gamma_0}^2} \tag{7a}$$

$$\delta C_Z(T, F_b) = \sqrt{\delta C_\Gamma^2 + \sigma_K^2 + \sigma_A^2} \tag{7b}$$

## 5 Sources of uncertainty and bias in Absolute Calibration with corner reflectors

In this section we identify and quantify the uncertainty and bias introduced by several terms in Eq. (2b). Following the recommendations in the work of Chandrasekar et al. (2015), we study the impact of receiver saturation, signal to clutter ratio, antenna lobe shape and antenna overlap. Additionally, we consider the impact of temperature fluctuations inside the radar box, loss changes with distance due to uneven amplification at the receiver IF and the effects of imperfect alignment of the reference target.

### 5.1 Receiver compression

It is advisable to design calibration experiments which avoid the appearance of compression effects. If this is not possible, compression must be considered in the data treatment so that the retrieved calibration remains valid in the receiver linear regime, where it usually operates during cloud sampling (Scolnik, 2000).

For studying how these effects could affect our calibration, we retrieved the radar receiver power transfer curve. Receiver characterization was done by removing the radar antennas and connecting the emitter end to the receiver input, with two attenuators in between. The first was a $40\ dB$ fixed attenuator, while the second was a tunable attenuator covering the range between $50$ and $1\ dB$ of losses. The adjustable attenuator enabled the retrieval of the power transfer curve by varying the attenuation and sampling the power at the receiver end (digital processing included). Our retrieved power transfer curve is shown in Fig. 4 (a).

Compression effects must be considered in calibration, or a bias will be introduced. In consequence, we include compression correction in every sample of reflected power, which consists on projecting their value to the ideal linear response using the power transfer curve.

For example, the power received from the 20 cm target on the 20 m mast returned was $4.1\ dBm$ in average, before corrections. The power transfer curve shows that at this power values we have a loss caused by compression of $\approx 0.3\ dB$. After correcting each power sample by compression with the power transfer curve, we obtain a corrected power average value of $4.5$ $dBm$. Meanwhile, for the 10 cm target on the 10 m mast the average power value before corrections is $3.2\ dBm$. As this value

is lower than what is obtained the 20 m mast, the associated compression effect is also smaller, of $\approx 0.2 \; dB$. After applying this correction to each power sample we end with a new corrected power average of $3.4 \; dBm$.

## 5.2 Antenna Properties

Manufacturer specifications indicate that antenna beamwidth should be of $0.8°$. However, data from an experimental characterization done by the same manufacturer in an anechoic chamber indicate that antenna beam shape is better approximated by a Gaussian with a Half Power Beam Width (HPBW) of $\theta \approx 0.88^o$. The total gain difference between the experimental curve and the Gaussian approximation of $\approx 0.0003 \; dB$ in the HPBW region. Therefore, we conclude that the contribution to uncertainty introduced by assuming a Gaussian beam shape is negligible. The Antenna beam shape and Gaussian curve are shown in Fig. 4 (b).

Another source of bias introduced by the antennas is the parallax error. Antenna parallax errors introduce a range dependent bias, determined by the antenna beamwidth and the relative angles of deviation between the antennas boresight. This bias is usually larger in the first few hundred meters closest to the radar. For example, for a deviation of half the antenna beamwidth, losses would be on the order of $10 \; dB$ and would vary significantly over the first hundreds of meters, decreasing with distance to about $1 \; dB$ at a approximately 4 kilometers (Sekelsky and Clothiaux, 2002).

To study this effect we took advantage of our experimental setup and the scanning capabilities of the radar, to check if the radar antennas were properly aligned. This was done by using the target on the 20 m mast. Results are shown in Fig. 4 (b). After analyzing the results we observed that the aiming uncertainty is in the same order of magnitude of the antennas beamwidth. Since the correction of the parallax error requires a very precise measurement of antenna alignment, we conclude that it is not possible to correct for antenna deviations directly with this information.

However, the relatively small difference of $0.5 \; dB$ in the estimation of $C_\Gamma^0$ during the calibration experiments of 2019, obtained using two masts in the most sensitive distance range (placed at $196$ and $376.5$ meters of distance respectively), indicate that antennas are unlikely to have a deviation comparable to their beamwidth (calibration results in Sect. 6).

Therefore, for the present version of this calibration methodology we assume that both antennas are parallel and that they have a Gaussian beam lobe. Once a reliable method for antenna pattern retrieval is developed for W band radars, it can be directly incorporated into the calibration term by adding an additional correction function $f_A(r)$ to Eq. (3). The uncertainty in this alignment estimation can also be included in the uncertainty budget, with the term $\sigma_A$ of Eq. (7b).

Even if antennas are parallel, it is necessary to include a correction for the loss $L_o(r)$ caused by incomplete antenna overlap. The correction, shown in Eq. (8), accounts for the loss of power that would be received from a point target compared to a monostatic system (Sekelsky and Clothiaux, 2002). This loss occurs because a point target cannot be in the center of two non-concentric parallel antenna beams.

$$L_o(r) = \exp\left( \frac{2\arctan(\frac{d}{2r})^2}{0.3606 \, \theta^2} \right) \tag{8}$$

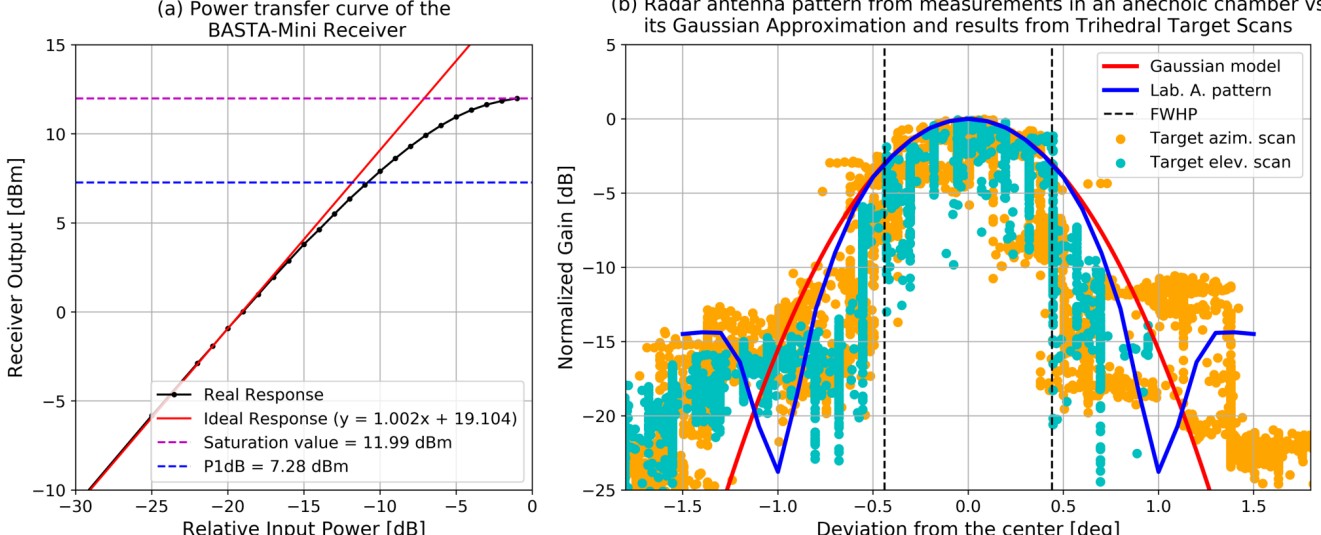

**Figure 4.** (a) Power transfer curve of the BASTA-Mini receiver. Input power is relative to the minimum attenuation value of the curve characterization experiment. All our signal retrievals from the target are slightly under the 5 [$dBm$] line, thus the correction required due to compression effects is small ($< 0.3\ dB$). (b) Normalized antenna pattern of the BASTA-Mini antennas. We can observe that the Gaussian fit with a beamwidth of $\theta = 0.88°$ is very close to the antenna gain curve measured at the manufacturer laboratories. This figure also shows the results from mast scans around the target to compare with the theoretical curves. To enable the comparison with the laboratory antenna pattern we assume that the gain of both antennas is identical. Then, the received power in $dBm$ is normalized with respect to the maximum measured value and divided by 2, to represent the gain of a single antenna.

Equation (8) assumes that the radar has two identical, parallel antennas with Gaussian beam lobes. Their main axis is separated by a distance $d$, and the point target is located at a distance $r$, facing the geometrical center of the radar, where the gain is maximum. For the BASTA-Mini $d = 35\ cm$. This introduces a loss of $0.08\ dB$ for the target at $r_0 = 196$ meters of distance, and of $0.02\ dB$ for the target at $r_0 = 376.5$ meters.

## 5.3 Signal to Clutter Ratio

The power sampled from our reference reflector is an addition of the power from the target (signal) and unwanted reflections on other elements in the environment, such as the ground or the mast (clutter). We observed that this clutter dominates above the radar noise, and thus becomes the main source of interference in our calibration signal.

To quantify the impact of clutter we use the Signal to Clutter Ratio (SCR) parameter. It is calculated as the ratio of total power received from the target to power received from clutter under the same configuration, but with the reference reflector removed. SCR enables the uncertainty $\sigma_{SCR}$ introduced by clutter in the sampled $P_r(r_0)$ values to be computed (Chandrasekar et al., 2015).

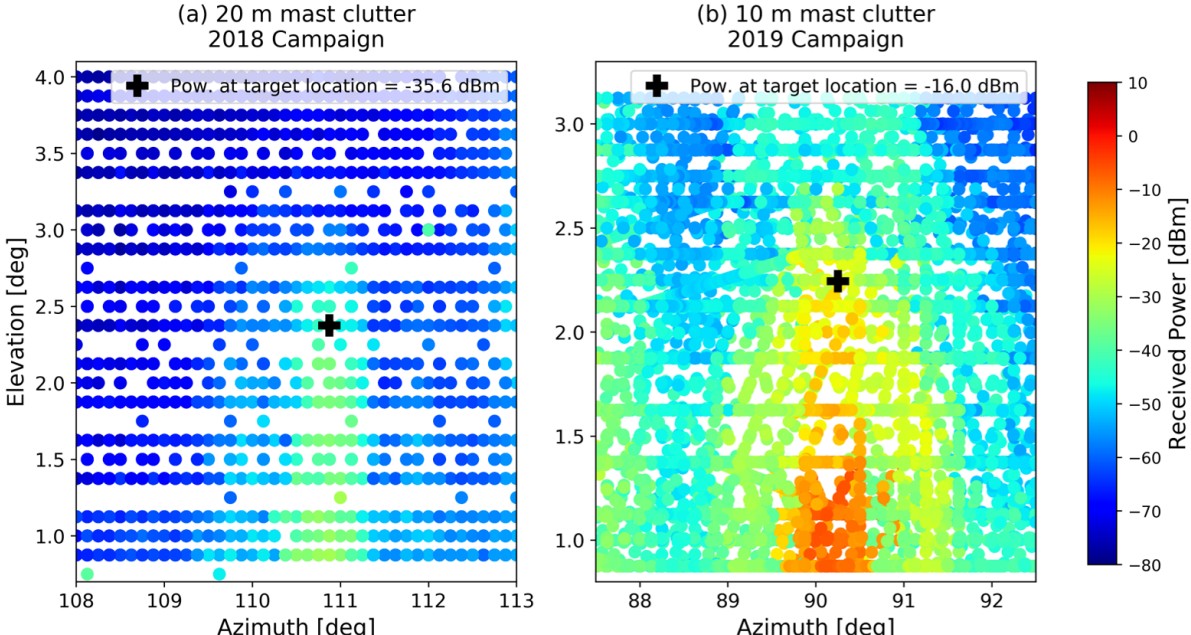

**Figure 5.** Clutter retrieval from the 10 m (a) and 20 m mast (b) respectively. Masts are scanned without the reflectors to measure the clutter signal. The nominal target position is marked with a black cross.

Clutter power is sampled and corrected following the same methodology used for reflector $P_r(r_0)$ retrievals, but in an scanning pattern mode to capture clutter around the mast area. Figure 5 shows our results from scanning around the 10 and 20 m masts with targets removed.

We observe that the 10 m mast is more reflective than the 20 m one. This may be caused by its smaller height (more ground clutter) and its larger geometrical cross-section. We can also see that the signal in the 10 m is stronger where absorbing material is not present (below $\approx 1.5°$ of elevation). In both cases we did not detect any signal from the nearby trees close to the target position.

To calculate SCR we compare the average power received from each target during the calibration experiments with the maximum clutter power observed in a region of $0.125^o$ around the target coordinates, vertically and horizontally. The value is taken from the radar scanner resolution.

The average power received from the 10 cm target on the 10 m mast is $3.4\ dBm$. This provides an SCR value of $19.4\ dB$, which implies a $\sigma_{SCR}$ uncertainty value of $\approx 0.93\ dB$. From the 20 cm target on the 20 m mast, the average received power is $4.5\ dBm$. Its SCR equals $40.1\ dB$, which is translated as an uncertainty contribution of $\sigma_{SCR} \approx 0.09\ dB$. From the results we see that even if target alignment is better with the 10 m mast, calibration results may not get less uncertain because the motor used for target alignment acts as a big source of clutter.

## 5.4 Temperature correction

BASTA-Mini has a regulation system to control temperature fluctuations inside the radar box. However, since the radar is based on solid state components, even small temperature fluctuations may impact the performance of the transmitter and receiver, and therefore affect the calibration stability. To account for this effect we introduced a temperature dependency in the calibration term, shown in Eq. (3).

During the experiments we verified the need of this correction by observing that the retrieved calibration term $C_\Gamma(T, F_0)$ has a consistent change depending on the time of the day, and that this change is strongly correlated to the temperature inside the radar.

Figure 6 (a), (b) and (c) show the results of a representative experiment done in the 2018 campaign. Here we left the radar sampling the target signal for several hours, to observe the variability of $C_\Gamma(T, F_0)$ during the day. (a) shows the raw result in the RCS calibration term $C_\Gamma(T, F_0)$. There is a spread of almost $1$ $dB$ between the maximum and minimum values during the whole timeseries. (b) is a Fourier transform of this raw timeseries. Here we can see that most of the variability happens in the timescale of hours. (c) presents the timeseries of (a), but in a daily cycle perspective. Here we plot hourly means of the deviation of $C_\Gamma(T, F_0)$ with respect to the total average, with its hourly standard deviation as errorbars. We also superimposed atmospheric attenuation and the radar amplifier temperature to show that the first has a much smaller impact in calibration variability compared to the second.

Figure (d) shows the raw results of plotting variations in $C_\Gamma(T, F_0)$ to temperature changes around $T_0 = 26.5$ $^oC$. These variations are calculated independently for each iteration, by subtracting the constant term of the linear fit of $C_\Gamma(T, F_0)$ with respect to temperature. This operation removes the effect introduced by differences in alignment between different iterations. The reference $T_0$ value is chosen because it is approximately the average internal temperature when considering all the experiments.

To maximize the range of temperatures covered we choose to not limit the sampling period to one hour. This decision has the drawback of increasing the noise of the dataset due to the inclusion of some data taken under suboptimal conditions, for example with wind speed velocities above 1 meter per second or with the presence of drizzle. Yet, this step is necessary to enable the retrieval of the temperature correction function for the widest range of temperatures possible.

To retrieve the temperature dependency we perform a linear regression over the results from all the experiments done in 2018 and 2019, shown in Fig 7. The regression shows that the variability in the calibration term has an almost linear relationship with internal radar temperature, in the $dB$ scale, and it is the same for both campaigns. This analysis allows us to estimate the value $n = 0.093$ $dB$ $^oC^{-1}$ for the temperature correction function of Eq. (3). To estimate the uncertainty of the temperature correction function we calculate the RMSE between the linear regression model and the whole dataset, for each degree of deviation in temperature. The RMSE value for the complete dataset is of $0.13$ $dB$, while its value per degree ranges between $0.07$ and $0.23$ $dB$ for a deviation of $0$ and $+3$ $^oC$ respectively. These results enable us to conclude that the temperature correction function uncertainty $\sigma_T$ is $\leq 0.23$ $dB$.

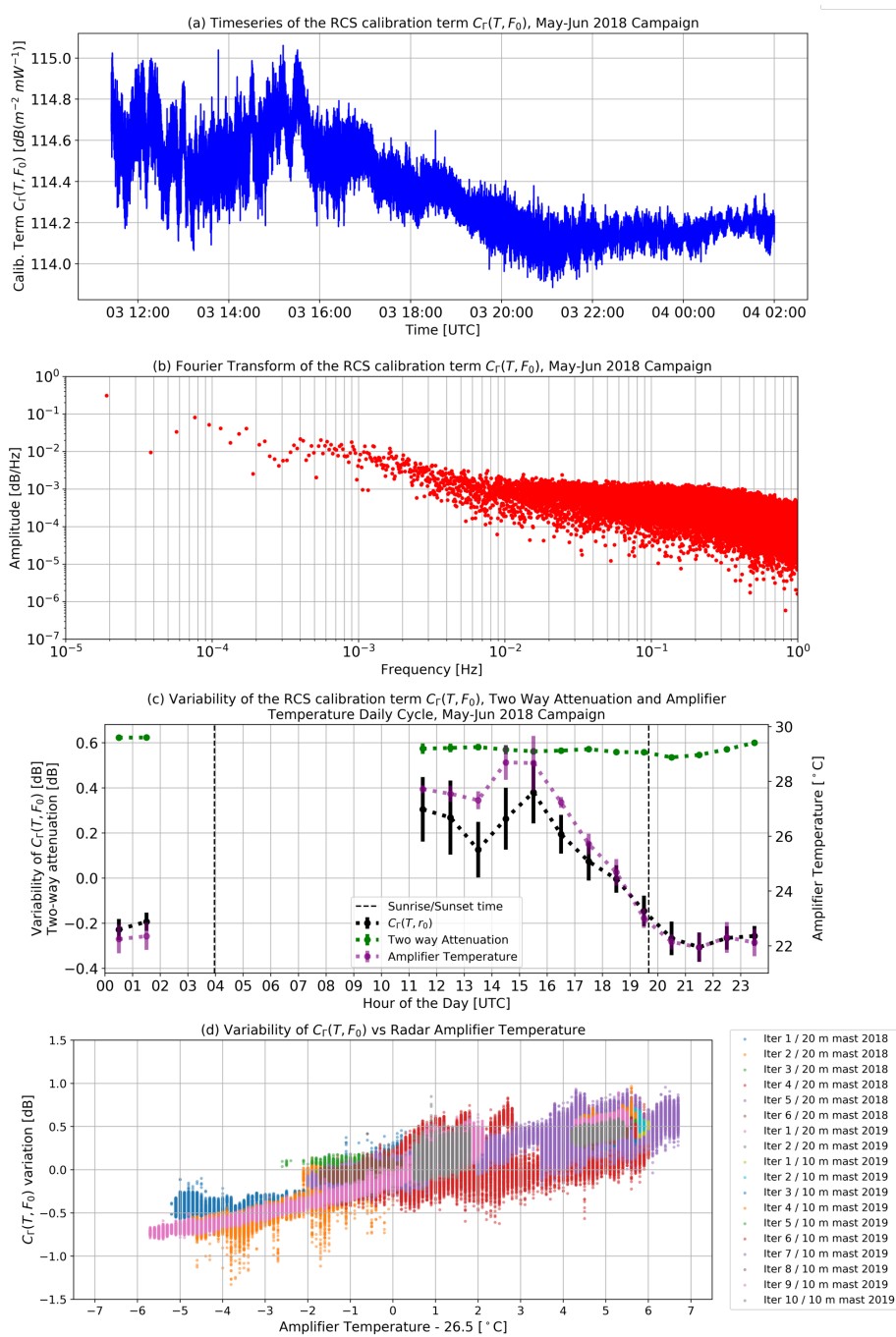

**Figure 6.** Calibration variability study. Samples from iteration 5, 2018 Calibration Campaign. (a) Time series of the RCS Calibration term retrieval. (b) Fourier transform of the RCS Calibration term after subtracting the mean value. (c) Calibration variability daily cycle, amplifier temperature and two-way attenuation. Attenuation errorbars are too small to be seen with this scale. (d) Relative changes in $C_\Gamma(T, F_0)$ versus amplifier temperature, plotted using all samples from 2018 and 2019 campaigns.

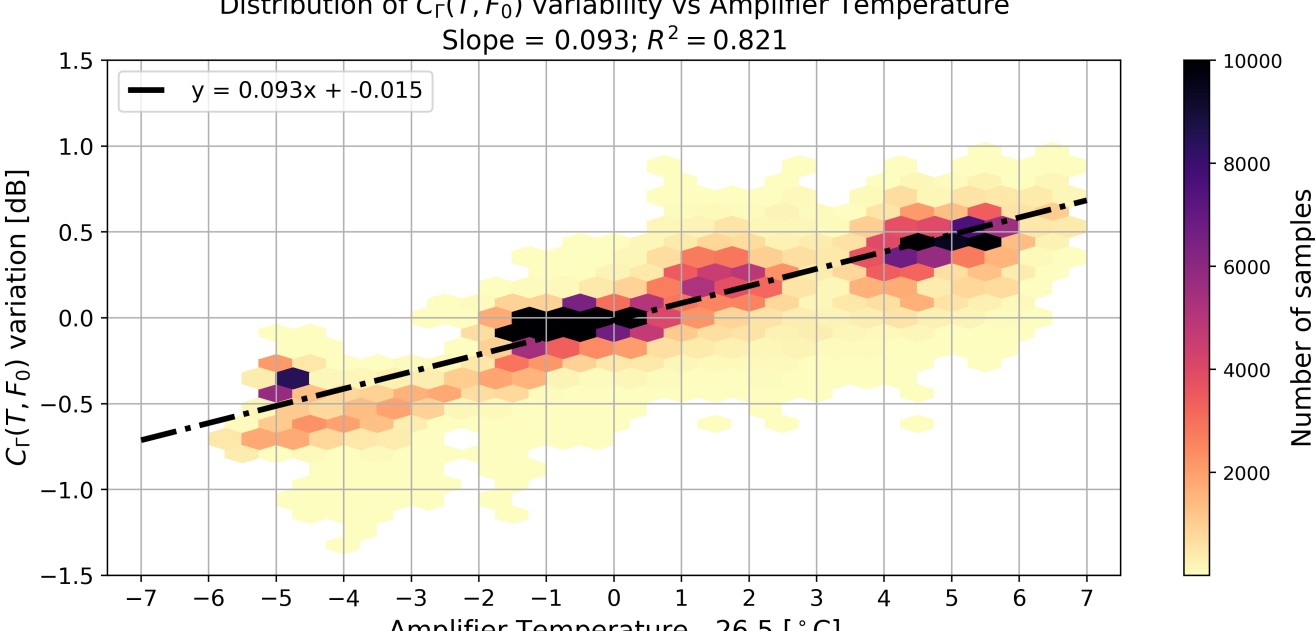

**Figure 7.** 2D histogram of the relative changes in $C_\Gamma(T, F_0)$ with respect to changes in the amplifier temperature and its linear least squares fit. The histogram is plotted using all $C_\Gamma(T, F_0)$ samples from 2018 and 2019 calibration campaigns.

## 5.5 IF loss correction function $f_{IF}(F_b)$

FMCW radars rely on estimating the beat frequency of the received signal to estimate the distance of an object. This signal
may suffer uneven amplification depending on its frequency, because of a frequency dependent gain function in the amplifiers of the IF chain of the radar. Since there is a direct relationship between the IF frequency $F_b$ and the target distance $r$, this dependency on the beat frequency introduces a gain variability with respect to the target distance $r$. As introduced in Sect. 2, this distance dependency is compensated in the calibration term with a IF correction function $f_{IF}(F_b)$.

The power $P_r(r)$ measured by the receiver when no active signal is input corresponds to the system noise power $N_s(F_b)$ plus
350 the environmental noise power $N_0$, amplified by the radar receiver gain $G_r(T, F_b)$ (this gain term is equivalent to $L_r^{-1}(T, F_b)$ of Eq. (2a)). Equation (9a) expresses this relationship when $P_r(r)$ is in $dBm$ and $N_0$, $N_s(F_b)$ are expressed in linear units (Pozar, 2009).

The standard way to retrieve each of these terms is to perform a two point calibration. This requires the use of two noise sources at significantly different and well known temperatures. Usually, the temperatures of the noise sources are the environ-
355 mental temperature (298 K) and that of liquid nitrogen (77 K) (Rodríguez Olivos, 2015). The receiver gain versus frequency retrieved from this two-point calibration could be used to derive the IF correction function directly. however, this approach

requires tailored equipment which was not available during the experimentation. Therefore, since the IF correction function is important to remove calibration bias, we follow a different approach to estimate its value.

To estimate the IF correction function we take advantage of the narrow IF bandwidth of the BASTA-Mini radar ($12\ MHz$, from 168 to $180\ MHz$). A calculation done with the Friis formula for the radar system indicates that the system noise $N_s(F_b)$ should have variations smaller than $0.1\ dB$ in this bandwidth. This can be explained by the large operating bandwidth and high gain of the receiver Low Noise Amplifier (LNA), of $35\ GHz$ and $> 20\ dB$ respectively, and by the small variation in the mixer conversion loss for the radar bandwidth ($< 0.3\ dB$). To verify the plausibility in the estimation of the noise figure variability, we performed an additional calculation testing the effect of varying the IF noise temperature from 0 to $400\ K$, and in all cases system noise variability remained under $0.1\ dB$.

This low variability enables the retrieval of the IF correction function by assuming a constant noise power density in the IF frequency range (Eq. (9b)). The constant noise power term $N_c$ corresponds to the addition of environmental and system noise.

$$P_r(r) \equiv P_r(F_b) = 10\ log_{10}\left(G_r(T, F_b) \cdot (N_s(F_b) + N_0)\right) \tag{9a}$$

$$\approx 10\ log_{10}\left(G_r(T, F_b) \cdot N_c\right) = 10\ log_{10}\left(\frac{N_c}{L_r(T, F_b)}\right) \tag{9b}$$

Then, to retrieve the $f_{IF}(F_b)$ we turn off the radar emitter and sample the environmental noise with the radar operating in its calibration configuration (12.5 meters distance resolution and 0.5 seconds integration time). After retrieving a significant amount of noise samples we calculate the average value of the difference $P_r(F_0) - P_r(F_b)$ for each IF frequency $F_b$, to remove the effect of the unknown noise power density. This operation is done to quantify relative gain variations around the calibrated frequency $F_0$.

By using Eqs. (2a) and (3), we get that the difference $P_r(F_0) - P_r(F_b)$ is equivalent to the difference between $C_\Gamma(T, F_b)$ and $C_\Gamma(T, F_0)$, and therefore it is equivalent to the IF correction function $f_{IF}(F_b)$ (Eq. (10)). The temperature effect in gain is removed because both $P_r(F_0)$ and $P_r(F_b)$ are sampled simultaneously, and therefore under the same temperature conditions.

$$\overline{P_r(F_0) - P_r(F_b)} = 10\log_{10}\left(\frac{L_r(T, F_b)}{L_r(T, F_0)}\right) = -C_\Gamma(T, F_0) + C_\Gamma(T, F_b) = f_{IF}(F_b) \tag{10}$$

For this experiment only, $P_r(F_0)$ corresponds to the power measured at the gate closer to the reference target position, without integrating other gates. This is done because there is no significant leakage and, as results of Fig. (8) show, $G_r(T, F_b)$ changes are negligible in the five gates used for integration.

Figure 8 shows the results of the IF correction function retrieval referenced to $P_r(F_0)$, using $F_0$ associated to the target distance $r_0 = 376.5$ meters (corresponding to the 20 m mast experiment setup). We can observe that all functions retrieved in 2019 are in close agreement, without significant variations between different dates or time of the day chosen for the plots. The 2018 function is different because hardware was modified between both calibration campaigns. Additionally, in 2018 the emitter was not turned off to perform the noise sampling. Rather, we resorted to use a sampling period with clear sky conditions

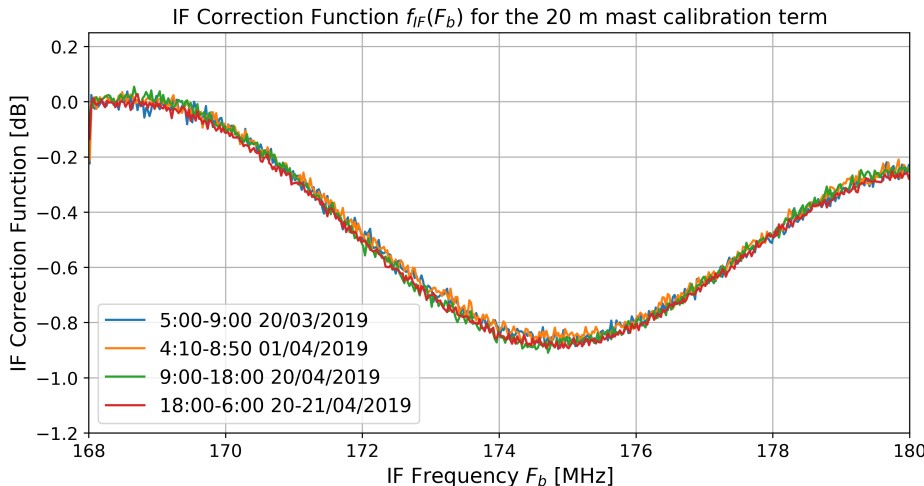

**Figure 8.** Data used for the IF correction function calculation, retrieved for different periods of the 2018 and 2019 calibration campaigns. 2018 curve is different from the 2019 results because hardware was modified between the campaigns. Time indicated in the label is in UTC.

to respect the assumptions of Eq. (9b). To avoid the effect of crosstalk, we only consider gates farther than 200 meters from the radar.

A sixth degree polynomial is used to fit $f_{IF}(F_b)$. For both 2018 and all 2019 curves, the fit has a RMSE $< 0.03 \ dB$. Furthermore, the standard deviation between results from the four periods of 2019 has a maximum value of $0.04 \ dB$ for any gate. Both results indicate that the uncertainty introduced by the IF correction function is $\leq 0.04 \ dB$. Finally, the IF correction function retrieved for the 10 m mast setup in 2019 (with $r_0 = 196$ meters) is almost identical to the 20 m mast results. These functions are presented in Sect. 6. Considering these low RMSE values, we decide to select as the IF correction function uncertainty the uncertainty introduced by assuming a constant system noise, thus $\sigma_{IF} = 0.1 \ dB$.

## 5.6 Misalignment Bias

The retrieval of $C_\Gamma(T, F_0)$ using Eq. (4b) requires a precise knowledge of the reference target effective RCS $\Gamma_0$. Each $dBsm$ of difference between the theoretical value used in calculations and the effective target RCS will introduce a bias of the same magnitude in the estimation of the calibration coefficient $C_\Gamma^0$, and thus in $C_\Gamma(T, F_0)$.

The effective reflector RCS is the actual physical value that would be measured by a perfectly calibrated radar. It is different from the target intrinsic RCS which only depends on its physical properties. Effective RCS changes when the experimental setup is modified. For example, if the point target is not exactly in the beam center, antenna gain will not be maximum and therefore the effective RCS will decrease compared to the intrinsic value. Effective RCS also changes when the incidence angle of the radar beam is modified. This latter effect may increase or decrease effective RCS depending on the original situation.

A common approach in these type of experiments is to set $\Gamma_0$ to be the maximum theoretical RCS of the target, assuming misalignment will cause a negligible deviation from this value. This procedure can be refined for cases where the system default

configuration does not have the target boresight aligned with the radar position. In these cases, effective RCS can be calculated using equations derived from geometrical optics (more complex optical calculations may be necessary for other wavelengths or target sizes). For example, we use the equations published by Doerry and Brock (2009) when calculating the effective RCS of our Triangular Trihedral target on the 20 m mast.

Unfortunately, this approach does not correct the impact of alignment uncertainties. We observed that random errors in the element positioning will statistically impact the effective $\Gamma_0$ in a single direction. Thus, simply taking the average of many target sampling iterations would result in a biased estimation of the calibration.

With the objective of quantifying the impact of alignment uncertainties we developed a geometrical simulator of effective RCS. This simulator receives as input the position of each element in the setup and calculates the effective RCS considering

the beam incidence angle and antenna gain variations when the target is not in the center of the beam. The degrees of freedom included in the simulator are shown in Fig. 9 (a). It enables the modification of the radar aiming angles, the mast dimensions and the positioning and orientation of the target. The equations used in the simulator can be found in the article support material.

We now use the simulator to study how uncertainty in alignment can affect the value of $\Gamma_0$. For this, we model an example experiment based on the 20 m mast setup. In this model we separate input variables between known and uncertain. Known

terms can be fixed or measured very precisely in the field experiment, hence they are set as fixed values. Meanwhile, uncertain terms represent the parameters that cannot be fixed or measured very precisely, and for that reason are better expressed as probability distributions (terms defined in Fig. 9 (a)).

- Known terms:

  - $x_r = 376.5$ m

- $h_r = 5.3$ m

  - $\rho = 20$ m

  - $\alpha = 48°$

  - Target Size = 20 cm

- Variables with uncertainty:

- $\theta_r = \mathcal{N}(\theta_r^*, \sigma_{\theta_r}^2)$

  - $\phi_r = \mathcal{N}(\phi_r^*, \sigma_{\phi_r}^2)$

  - $\theta = \mathcal{N}(0, \sigma_\theta^2)$

  - $\phi = \mathcal{U}([0°, 360°))$

  - $\tau = \mathcal{N}(\tau^*, \sigma_\tau^2)$

In the uncertain variables, $\theta_r^* = 87.82°$, $\phi_r^* = 0°$ and $\tau^* = 0°$ represent the nominal alignment angles, which are the values expected under an ideal field experiment where the radar aims directly to the target and the mast is perfectly vertical. To these

nominal values we associate a distribution shape and the uncertainty set $\sigma_{\theta_r} = 0.075°, \sigma_{\phi_r} = 0.075°, \sigma_\theta = 1.5°, \sigma_\tau = 5°$. Each term, known and uncertain, is estimated from observations done during the experimental field work.

With these input parameters we sample the $\Gamma_0$ distribution that would arise after a large amount of experimental iterations. Figure 9 (b) shows the results from this sampling. The black dashed line shows the effective RCS under our experimental configuration, when each element is in its nominal position. We can see that this effect cannot be neglected in our case, since its value is $0.8\ dB$ lower than the maximum theoretical RCS.

However, this single correction does not suffice. The results of the model show that the addition of uncertainty into the process induces another bias of $\approx 0.3\ dB$, in average. Since this is withing the order of magnitude of our desired uncertainty in the calibration, the example clearly illustrates the need of including a bias correction step in our calibration methodology.

The standard deviation $\sigma_\epsilon$ between $N$ experimental retrievals of $C_{\Gamma i}^0$ cannot be used directly as an estimation of uncertainty because the RCS distribution shape is not Gaussian. The uncertainty introduced by this variability is studied by sampling a large set of possible RCS distributions based on our experimental configuration, and selecting the candidates matching our observed spread $\sigma_\epsilon$. This set provides an estimation of the expected bias correction $\tilde{\Lambda}$ and of the effective RCS uncertainty $\sigma_\Lambda$. The uncertainty of the $C_\Gamma^0$ estimator of Eq. (6) will correspond to the uncertainty of each $C_{\Gamma i}^0$ estimation propagated through the calculation of their average (terms $\sum \sigma_i^2/N^2$ and $\sigma_T^2/N$ of Eq. (7a)) plus the effective RCS uncertainty $\sigma_\Lambda$. The details on how this estimator works and how the RCS distribution sampling is done are fully explained in Sect. S3 of the supplementary material.

## 6   Results

In 2018 we used the 20 m mast only, performing six iterations. For 2019 we did 10 iterations using the 10 m mast and 2 iterations with the 20 m mast. The distributions of $C_\Gamma^0$ obtained in each iteration and experiment is shown in Fig. 10.

The radar hardware changed between 2018 and 2019 campaigns due to experiments required to retrieve the power transfer curve and perform maintenance operations. This implies that we cannot compare absolute calibration values between both campaigns. What remains valid is to compare properties such as the variability, and the results from both experiments of 2019.

In the results we can notice a difference in $C_{\Gamma i}^0$ spread when comparing the 10 and 20 m masts. The 6 iterations of 2018 (Fig. 10 (A)) have an spread of $\sigma_\epsilon = 0.33\ dB$, while the spread of the 10 iterations of 2019 is $0.11\ dB$ (Fig. 10 (B)). This happens because the 10 m mast has a motor on top which enables a much finer adjustment of the target position, improving the repeatability of the experiments.

There is also a small difference in the spread of the curves. The $C_{\Gamma i}^0$ values retrieved in experiment (B) have a smaller spread $\sigma_i$. This is because we took all the samples during one single night, with very clear conditions and an average wind speed below 1 $m/s$. A great advantage was the presence of the motor that enables target alignment in $\approx 5$ minutes. Meanwhile, for experiment (A) curves were sampled during different days, because the 20 m mast setup requires more time to align ($\approx 2$ hours). The different conditions in each day led to a more varied shape in the retrieved curves. This effect is specially noticeable in experiment (C), where the iterations were performed during daytime, when atmospheric conditions are more

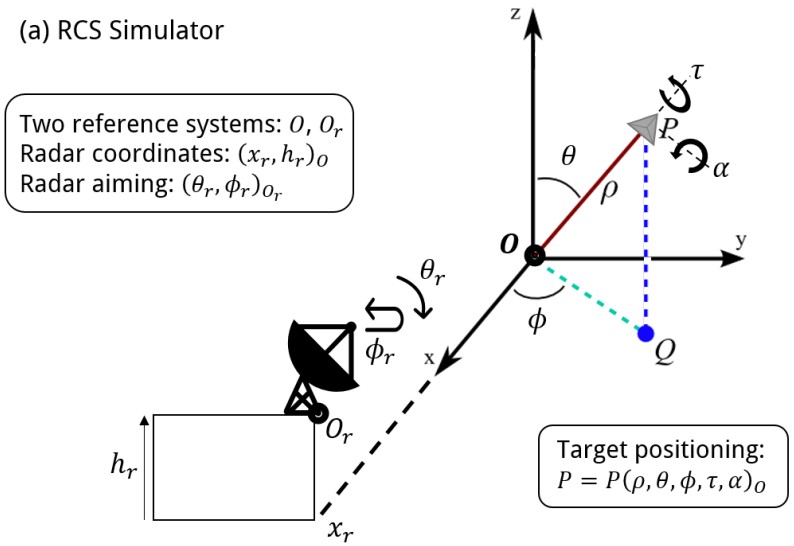

(a) RCS Simulator

Two reference systems: $O$, $O_r$
Radar coordinates: $(x_r, h_r)_O$
Radar aiming: $(\theta_r, \phi_r)_{O_r}$

Target positioning:
$P = P(\rho, \theta, \phi, \tau, \alpha)_O$

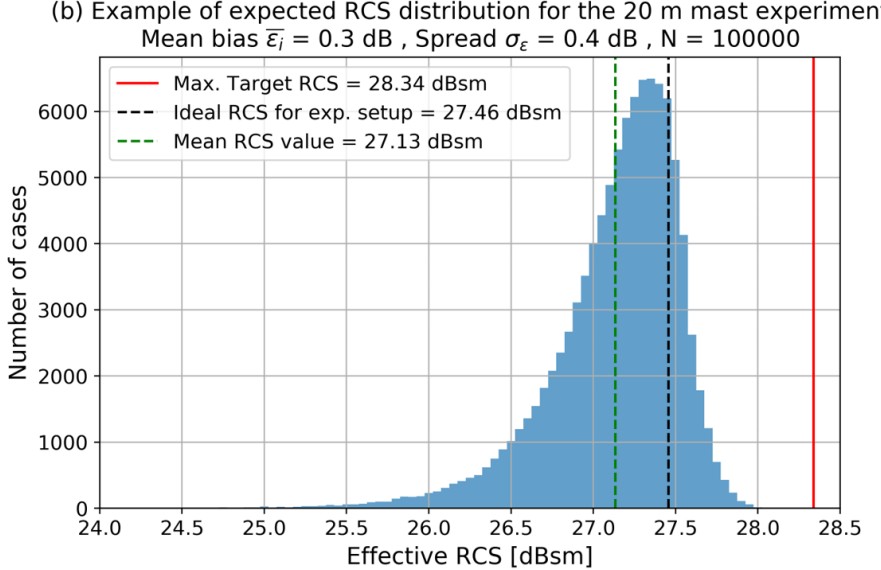

(b) Example of expected RCS distribution for the 20 m mast experiment
Mean bias $\overline{\varepsilon}_i = 0.3$ dB , Spread $\sigma_\varepsilon = 0.4$ dB , N = 100000

— Max. Target RCS = 28.34 dBsm
--- Ideal RCS for exp. setup = 27.46 dBsm
--- Mean RCS value = 27.13 dBsm

**Figure 9.** (a) Diagram of the RCS simulator illustrating its degrees of freedom. (b) Example of an effective RCS distribution obtained after 100 000 simulations with the uncertainty set specified in the text. The simulations are based on our 20 m mast setup. Bias is calculated subtracting the ideal RCS by the mean RCS value. The example illustrates how the effective RCS will be, statistically, lower than the result expected from an ideally aligned setup.

dynamic, specially wind speed variability. The introduced variability was not fully compensated by our corrections and thus bimodal distributions remained. However, individual spread is still small, within $\approx 0.1$ dB, so we decided to accept these samples for calibration purposes.

To study the dependency of the bias correction on the amount of iterations we calculate the bias correction term $\tilde{\Lambda}$ and its uncertainty $\sigma_\Lambda$ of experiments (A) and (B) with different amounts of repetitions. The order of the iterations used in each row

match the sequential order indicated in Fig. 10. The results are shown in Table 1. For both cases we have the best estimate when we use all the samples available for each experiment, and thus we use this bias correction and uncertainty when computing the calibration coefficient.

For experiment (C) we followed a different approach. Because we only have two samples, the calculated $\sigma_\epsilon = 0.2\ dB$ is very likely to be underestimated. Consequently, and because the experimental procedure was identical to what was done in

2018, we assume our parameters $\sigma_\epsilon$, $\tilde{\Lambda}$ and $\sigma_\Lambda$ to be equal to the best estimation of experiment (A). This is possible because in our methodology we assume that the bias probability distribution of a given system is unique, even if it is unknown, and what is done by performing many iterations is to successively restrict the possible sets of uncertainties that can generate results consistent with the observations. This latter hypothesis is consistent with the decrease in uncertainty for the bias correction when increasing the amount of iterations.

Table 2 contains a summary of all known bias corrections and uncertainty contributions, introduced in Sect. 4. With the aforementioned results, we use Eqs. (6), (3), (7a) and (7b) to estimate the RCS and Reflectivity calibration terms $C_\Gamma(T, F_b)$ and $C_Z(T, F_b)$, alongside their uncertainty. Since the term $\sigma_{\Gamma_0}$ is much larger than all other uncertainty sources, we calculate a partial calibration uncertainty including all but this term, to simplify the comparison of uncertainty contributions between different experimental setups. This term is then added for the calculation of the final result. $C_Z(T, F_b)$ is calculated for the

range resolution $\delta r = 12.5\ m$, which is the same mode used for target sampling. $T$ is the radar amplifier temperature in $^oC$ and $f_{IF}(F_b)$ is the IF loss correction function.

- (A) 20 m mast - 2018:

  ⋄ $C_\Gamma(T, F_b) = -80.98 + 0.093(T - 26.5) + f_{IF}(F_b)\ [dB(m^{-2}\ mW^{-1})] \pm 2\ [dB]$

  ⋄ $C_Z(T, F_b) = 3.05 + 0.093(T - 26.5) + f_{IF}(F_b)\ [dB(mm^6\ m^{-5}\ mW^{-1})] \pm 2\ [dB]$

⋄ $f_{IF}(F_b) = 7.34 \cdot 10^{-6} F_b^6 - 7.70 \cdot 10^{-3} F_b^5 + 3.36 F_b^4 - 7.83 \cdot 10^2 F_b^3$
  $+ 1.02 \cdot 10^5 F_b^2 - 7.15 \cdot 10^6 F_b + 2.08 \cdot 10^8\ [dB]$

- (B) 10 m mast - 2019:

  ⋄ $C_\Gamma(T, F_b) = -79.76 + 0.093(T - 26.5) + f_{IF}(F_b)\ [dB(m^{-2}\ mW^{-1})] \pm 2\ [dB]$

  ⋄ $C_Z(T, F_b) = 4.28 + 0.093(T - 26.5) + f_{IF}(F_b)\ [dB(mm^6\ m^{-5}\ mW^{-1})] \pm 2\ [dB]$

⋄ $f_{IF}(F_b) = 7.60 \cdot 10^{-6} F_b^6 - 7.97 \cdot 10^{-3} F_b^5 + 3.48 F_b^4 - 8.10 \cdot 10^2 F_b^3$
  $+ 1.06 \cdot 10^5 F_b^2 - 7.40 \cdot 10^6 F_b + 2.15 \cdot 10^8\ [dB]$

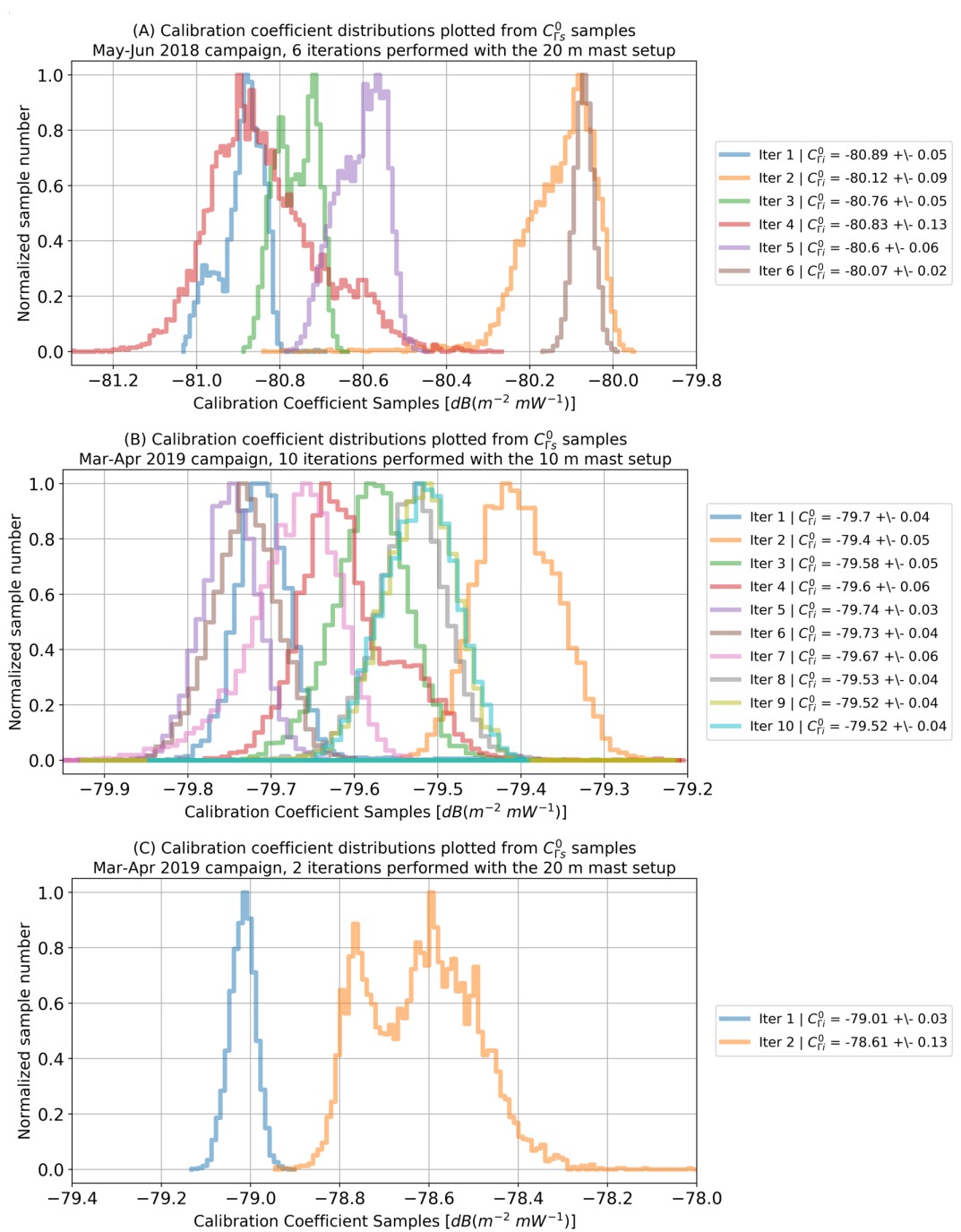

**Figure 10.** Calibration coefficient distributions obtained for (A) 2018 campaign using the 20 cm target on the 20 m mast, (B) 2019 campaign using the 10 cm target on the 10 m mast and (C) 2019 campaign with the 20 cm target on the 20 m mast.

**Table 1.** Bias correction $\tilde{\Lambda}$ and its uncertainty $\sigma_\Lambda$ calculated using a different amount of iterations, for the experiments of 2018 and 2019 calibration campaigns (for ex. 3 iterations means we used iterations 1, 2 and 3 of the experiment). We include the average and spread $\sigma_\epsilon$ between the retrieved $C_{\Gamma i}^0$ for each case. This variability $\sigma_\epsilon$ is introduced in the bias estimation procedure to determine the bias correction $\tilde{\Lambda}$ and its uncertainty $\sigma_\Lambda$.

| (A) 20 m mast 2018 | Exp. Results | | Bias Correction | |
| --- | --- | --- | --- | --- |
| N$^o$ of iterations | $\frac{1}{N}\sum C_{\Gamma i}^0$ | $\sigma_\epsilon$ [dB] | $\tilde{\Lambda}$ [dB] | $\sigma_\Lambda$ [dB] |
| 2 | -80.51 | 0.38 | 0.98 | 1.78 |
| 3 | -80.59 | 0.33 | 0.65 | 0.86 |
| 4 | -80.65 | 0.31 | 0.51 | 0.50 |
| 5 | -80.64 | 0.28 | 0.40 | 0.33 |
| 6 | -80.54 | 0.33 | 0.44 | 0.28 |
| (B) 10 m mast 2019 | | | | |
| N$^o$ of iterations | | | | |
| 2 | -79.55 | 0.15 | 0.78 | 1.65 |
| 3 | -79.56 | 0.12 | 0.42 | 0.70 |
| 4 | -79.57 | 0.11 | 0.27 | 0.34 |
| 5 | -79.60 | 0.12 | 0.24 | 0.20 |
| 6 | -79.62 | 0.12 | 0.22 | 0.13 |
| 7 | -79.63 | 0.11 | 0.19 | 0.10 |
| 8 | -79.62 | 0.11 | 0.18 | 0.07 |
| 9 | -79.61 | 0.11 | 0.17 | 0.06 |
| 10 | -79.60 | 0.11 | 0.16 | 0.05 |
| (C) 20 m mast 2019 | | | | |
| N$^o$ of iterations | | | | |
| 2 | -78.81 | - | 0.44 | 0.28 |

- (C) 20 m mast - 2019:

  ◇ $C_\Gamma(T, F_b) = -79.25 + 0.093(T - 26.5) + f_{IF}(F_b)\ [dB(m^{-2}\ mW^{-1})] \pm 2\ [dB]$

  ◇ $C_Z(T, r) = 4.79 + 0.093(T - 26.5) + f_{IF}(F_b)\ [dB(mm^6\ m^{-5}\ mW^{-1})] \pm 2\ [dB]$

  ◇ $f_{IF}(F_b) = 7.60 \cdot 10^{-6} F_b^6 - 7.97 \cdot 10^{-3} F_b^5 + 3.48 F_b^4 - 8.10 \cdot 10^2 F_b^3$
  $+ 1.06 \cdot 10^5 F_b^2 - 7.40 \cdot 10^6 F_b + 2.15 \cdot 10^8\ [dB]$

These results enable the analysis of relative uncertainty contributions from different sources, however the total calibration uncertainty may be underestimated. As indicated in Sects. 4 and 5, some bias terms remain unknown. Specifically, target physical RCS must be measured in an echo chamber to improve the misalignment bias estimation. In addition, the method to characterize antenna alignment must be improved to determine if there is a need for an additional distance correction function

**Table 2.** Summary of all known corrections and uncertainty contributions in the calculation of $C_\Gamma(T, F_b)$. The absolute correction terms have a sign associated with the direction in which they impact the final calibration calculation. For the receiver compression correction we present the average magnitude and for the temperature correction we present the range of possible values. The Partial Calibration Uncertainty is the addition of all uncertainty terms except $\sigma_{\Gamma_0}$. This term is later added to calculate the Total Calibration Uncertainty.

| Absolute Corrections | Term [dB] | (A) 20 m mast 2018 | (B) 10 m mast 2019 | (C) 20 m mast 2019 |
|---|---|---|---|---|
| Compression | Fig. 4 (a) | -0.3 in avg. | -0.2 in avg. | -0.3 in avg. |
| Partial Antenna Overlap | $L_o(r_0)$ | -0.02 | -0.08 | -0.02 |
| Temp. Corr. ($T_0 = 26.5\ °C$) | $n(T - T_0)$ | within $\pm 0.6$ | within $\pm 0.6$ | within $\pm 0.6$ |
| Misalignment Bias | $\tilde{\Lambda}$ | -0.44 | -0.16 | -0.44 |
| IF loss correction | $f_{IF}(F_b)$ | $\leq |0.6|$ | $\leq |0.9|$ | $\leq |0.9|$ |
| **Uncertainty Sources** | **Term [dB]** | | | |
| $C_{\Gamma i}^0$ estimation | $\sqrt{\frac{1}{N^2} \sum \sigma_i^2}$ | 0.03 | 0.01 | 0.07 |
| Temp. Corr. in $C_{\Gamma i}^0$ retrievals | $\frac{\sigma_T}{\sqrt{N}}$ | 0.09 | 0.07 | 0.16 |
| Temp. Corr. in $C_\Gamma(T, F_b), C_Z(T, F_b)$ | $\sigma_T$ | 0.23 | 0.23 | 0.23 |
| Signal to Clutter Ratio | $\sigma_{SCR}$ | 0.09 | 0.93 | 0.09 |
| Bias Correction | $\sigma_\Lambda$ | 0.28 | 0.05 | 0.28 |
| IF loss correction | $\sigma_{IF}$ | 0.1 | 0.1 | 0.1 |
| Partial Calibration Uncertainty | | 0.40 | 0.97 | 0.43 |
| Reflector RCS Uncertainty | $\sigma_{\Gamma_0}$ | 2 | 2 | 2 |
| Total Calibration Uncertainty | $\delta C_\Gamma; \delta C_Z$ | 2.04 | 2.22 | 2.04 |

(Sect. 5.2). The uncertainty of these retrievals will impact the total uncertainty value, however, it is possible to quantify this effect through the terms $\sigma_{\Gamma_0}$ and $\sigma_A$ of Eq. (7b).

To finalize, we perform a test of the calibration results by measuring a altostratus cloud in both campaigns (Fig. 11). The sampling was done with the $25\ m$ resolution, and thus $6\ dB$ had to be subtracted from the $C_Z(T, F_b)$ calibration calculated for the $12.5\ m$ resolution. In this correction, $3\ dB$ come from the change in the distance resolution term $\delta r$ (Eq. (5a)), and the other $3\ dB$ are subtracted to compensate the additional digital gain coming from doubling the amount of points in the chirp Fourier transform (Delanoë et al., 2016). A Signal to Noise Ratio threshold of $8\ dB$ is used to remove noise samples. We observe that for both campaigns the reflectivity measured in altostratus cloud is within $-30$ - $0\ dBZ$, which are typical values reported in literature (Uttal and Kropfli, 2001).

## 7 Conclusions

This study presents a cloud radar calibration method that is based on cloud radar power signal backscattered from a reference reflector. We study the validity of the method and variability of the results by performing measurements in two experimental setups and analyzing the associated results. In the first experimental setup we use a scanning BASTA-Mini W-band cloud radar, that aims towards a 20-cm triangular trihedral target installed at the top of a 20-m mast, located 376.5 m from the radar. For the second experimental setup, we use the same radar, aimed towards a 10-cm triangular trihedral target mounted on a pan–tilt motor at the top of a 10-m mast. The mast is located 196 m from the radar.

The first consideration in the design of the experimental setup is the need to avoid excessive compression or saturation in the radar receiver. This must be checked before any calibration attempt by comparing measurements of radar backscattered power with the radar receiver power transfer curve. In both our setups we find losses due to compression on the order of $0.2 \sim 0.3$ $dB$. There is a compensating effect between target RCS and radar-to-target distance (Eq. 2b). Since the compression effect is small, we correct it using our receiver power transfer curve. However, in cases where the radar is operating close to saturation, or when compression effects are larger than the calibration uncertainty goal, it is advisable to compensate by reducing target size or by positioning the target farther away from the radar.

Secondly, the reflector must be positioned far enough from the radar to be outside the antennas near-field distance and to ensure that the received power has low antenna-overlap losses. The BASTA-Mini cloud radar has a Fraunhofer near-field distance of $50$ m. The estimated maximum overlap loss is less than $0.1\ dB$ for the closest (10-m) mast setup. Thus we conclude that the target positioning is far enough for both setups.

Thirdly, the experimental setup should strive to reduce clutter in the radar measurements. This can be achieved by operating in an open field that is several hundred meters in length and free of trees or other signal-inducing obstacles. It is also advisable to perform radar measurements under clear conditions, without fog or rain, with wind speed below $1\ ms^{-1}$ and low turbulence.

Next, the proposed calibration method requires performing several iterations in the same setup configuration. In each iteration the setup is first realigned, followed by approximately one hour of sampling of the reference reflector backscattered power. The sampled power is then corrected for compression effects, incomplete antenna overlap, variations in radar gain due

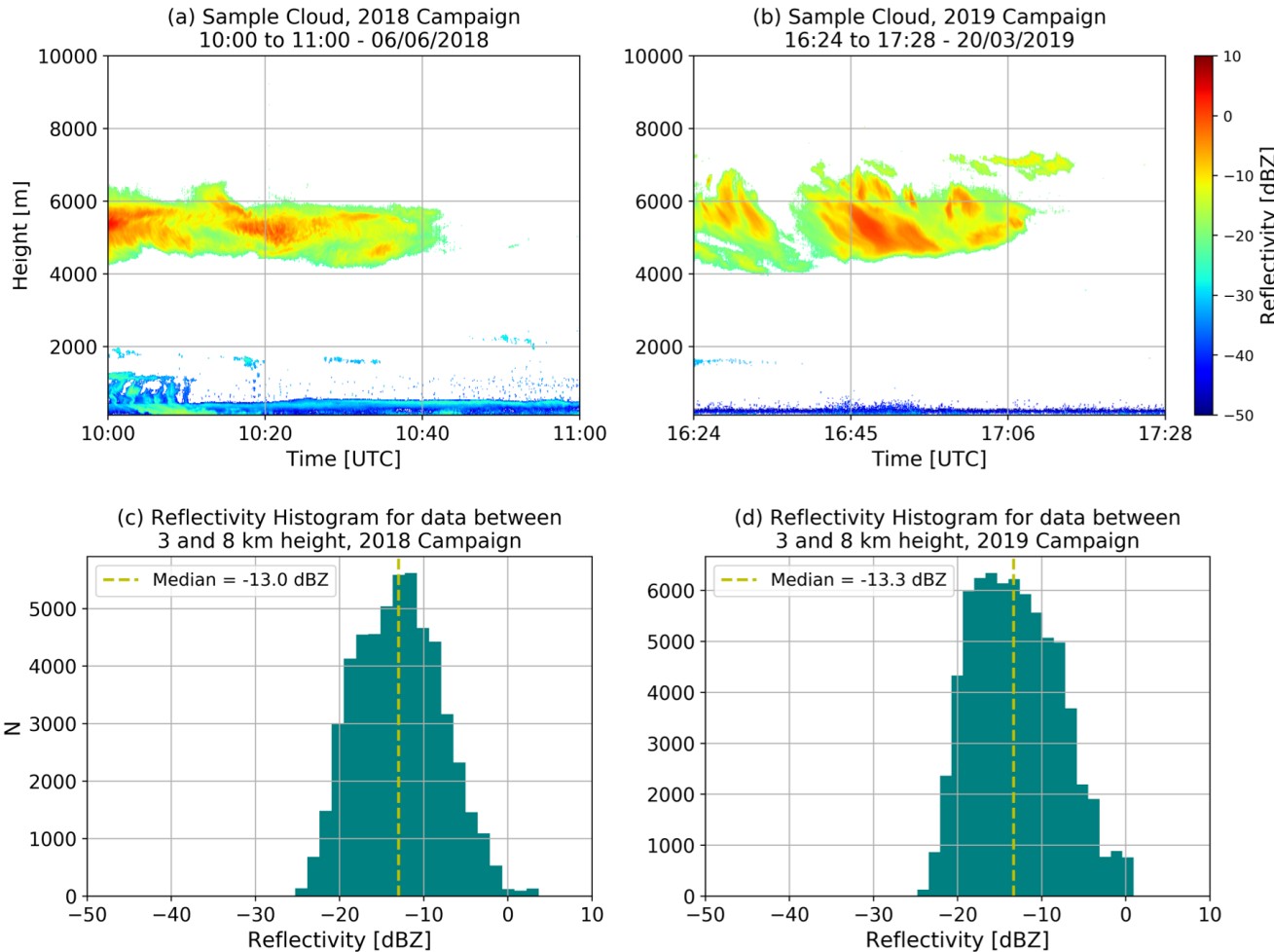

**Figure 11.** Altostratus cloud sampled during 2018 (a) and 2019 campaigns (b). Lower reflectivities are easier to capture at lower altitudes because of lower distance and attenuation losses (Eq. (5b)). In the altostratus reflectivity histograms (c) and (d) we observe that for both campaigns measurements are within the ranges reported in literature.

to temperature and atmospheric attenuation, before being used to estimate a RCS calibration term value. Once all iterations are completed, the final RCS and Equivalent Reflectivity calibration terms can be computed with their respective uncertainties.

Iterations are necessary because they enable the quantification of bias introduced by inevitable system misalignment. Our experiments indicate that, for our setup, at least 5 iterations are necessary to reach convergence in the calculation of bias and uncertainty associated with misalignment. We find a bias correction of $\approx 0.4 \pm 0.3\ dB$ for the 20-m mast, and of $\approx 0.2 \pm 0.1\ dB$ for the 10-m mast. This difference can be explained by the more precise alignment attainable with the pan–tilt motor installed on the 10 m mast.

Calibration is also impacted by changes in the gain of radar components associated with internal temperature variations. For the radar used in our experiment, these changes reach up to $\pm 0.6\ dB$. Our experiments enabled us to retrieve a correction function for the temperature dependence and to reduce the temperature uncertainty contribution to $\sigma_T = 0.23\ dB$. This result indicates that lower calibration uncertainties can be achieved by studying temperature effects, especially for solid state radars.

Another necessary consideration is the inclusion of gain variations with distance, introduced by frequency dependent losses in the IF of the radar receiver. We found calibration variations with distance up to $0.9\ dB$ for the 2019 campaign. Therefore, characterizing the IF loss is a necessary step to validate the calibration results for all ranges.

Our analyses reveal that the predominant source of uncertainty for all experiments is the reference target RCS, reaching approximately $2\ dB$ due to the use of a theoretical model instead of an experimental characterization. The next most important contributions to uncertainty come from the levels of clutter and alignment precision. These two effects have different magnitudes in our two experimental setups (10-m and 20-m masts). The 20-m mast setup uncertainty is limited by the uncertainty contribution of the alignment bias estimation $\sigma_\Lambda = 0.28\ dB$. The 10-m mast setup uncertainty is limited by the uncertainty contribution of the signal-to-clutter ratio $\sigma_{SCR} = 0.9\ dB$. This result reveals that there is a tradeoff between better target alignment and additional clutter introduced by the alignment motor.

The complete uncertainty budget enables us to conclude that to reach a calibration uncertainty under $1\ dB$, it is necessary to have a target RCS characterization with an uncertainty lower than $0.9\ dB$, based on the accumulated uncertainty of all terms, except target RCS, of $0.4\ dB$. This uncertainty was obtained using the 20-cm target on the 20 m mast during the 2018 experiment, where six target sampling iterations were performed.

Finally, because of cloud radar hardware modifications in the fall of 2018, the calibration coefficients found in May 2018 and March 2019 differ by $1.2\ dB$. We compare cloud radar measurements of altrostratus clouds performed in May 2018 and March 2019. The reflectivity distributions of the two events are consistent and compatible with values previously registered in literature. The two distributions yield median values that differ by $0.3\ dB$.

For future work we envisage to develop a technological solution to allow target orientation without introducing additional clutter. Another interesting prospect is to improve the accuracy of the radar scanner, to enable direct retrieval of antenna pattern directly with the radar, following the method proposed by Garthwaite et al. (2015). This retrieval would improve the bias correction arising from parallax errors, which at present is calculated assuming parallel radar antennas.

We also plan to perform a receiver noise figure characterization, to further reduce uncertainty in the IF correction, and an echo chamber characterization of our reference targets. Target characterization will enable the removal of bias caused by

manufacturing imprecision, reduce the RCS uncertainty contribution to total uncertainty and improve the estimation of our
system misalignment bias correction.

Further, there is ongoing research on calibration and antenna pattern characterization methods based on reference targets held by Unmanned Aerial Vehicles (UAVs) (Duthoit et al., 2017; Yin et al., 2019). Since the underlying principle is the same, most considerations written here should be directly applicable in these new experiments. Here the UAV takes the role of the mast, holding the reflector (usually a sphere), and therefore it is important to characterize the UAV RCS and verify that it
does not interfere with the experiment. The main difference would be in the procedure necessary to estimate bias, because the reference target (usually a sphere) will be always moving due to wind. Here an adaptation of the effective RCS simulator would be necessary to account for the target type and different alignment protocol.

*Data availability.*

All data used in this study is hosted by the SIRTA observatory. Data access can be requested for free following the conditions
indicated in the SIRTA data policy (https://sirta.ipsl.fr/data_policy.html).

SIRTA observatory website: https://sirta.ipsl.fr/

Data request form: https://sirta.ipsl.fr/data_form.html

*Author contributions.*

All authors contributed to the planning of the campaigns and the design of the calibration experiments.
Author Julien Delanoë was responsible of radar installation and operation.

Authors Jean-Charles Dupont and Felipe Toledo worked in the preparation, development and operation of the necessary infrastructure for the experiments.

Authors Julien Delanoë and Felipe Toledo retrieved the Power Transfer Curve of the Radar Receiver.

Data analysis and the establishment of the calibration methodology presented in the paper was done by Felipe Toledo.
Authors Martial Haeffelin and Felipe Toledo worked in defining the paper structure and content.

Authors Felipe Toledo, Susana Jorquera and Cristophe Le Gac worked in developing the method to retrieve the IF correction function, and in its calculation.

Author Christophe Le Gac contributed with technical information about the radar.

All authors reviewed the paper.

*Competing interests.*

Author Felipe Toledo has received research funding from Company Meteomodem.

*Acknowledgements.* The authors would like to acknowledge Johan Parra, Patricia Delville, Cristophe Boitel and Marc-Antoine Drouin from the SIRTA Atmospheric Observatory for their assistance in the execution of the field experiments. This acknowledgment is extended to Razvan Pirloaga and Dragos Ene from the INOE Institute, Romania. We would also like to thank Fabrice Bertrand and Jean-Paul Vinson from the LATMOS Laboratory, France, for their collaboration.

We would like to acknowledge the two reviewers for their expert comments which enabled us to improve the proposed calibration method.

We also acknowledge the french *Association Nationale de la Recherche* (ANRT) and the company Meteomodem for their contribution in the funding of this work. Finally, we state this work is part of the ACTRIS-2 project and has received funding from the European Union's Horizon 2020 research and innovation programme under grant agreement No 654109.

## Table of Symbols

| Symbol | Description | Units |
|---|---|---|
| $C_\Gamma(T, F_b)$ | RCS Calibration Term | $dB(m^{-2}\ mW^{-1})$ |
| $C_\Gamma(T, F_0)$ | RCS Calibration Term at the IF frequency $F_0$ | $dB(m^{-2}\ mW^{-1})$ |
| $C_\Gamma^0$ | RCS Calibration Coefficient | $dB(m^{-2}\ mW^{-1})$ |
| $C_{\Gamma s}^0$ | Single sample of the Calibration Coefficient $C_\Gamma^0$ | $dB(m^{-2}\ mW^{-1})$ |
| $C_{\Gamma i}^0$ | Mean value of all $C_{\Gamma s}^0$ samples retrieved in iteration $i$, | $dB(m^{-2}\ mW^{-1})$ |
| $C_Z(T, F_b)$ | Radar Equivalent Reflectivity Calibration Term | $dB(mm^6\ m^{-5}\ mW^{-1})$ |
| $\delta C_\Gamma$ | RCS Calibration Uncertainty | $dB$ |
| $\delta C_Z$ | Radar Equivalent Reflectivity Calibration Uncertainty | $dB$ |
| $F_b$ | Signal frequency at the radar receiver IF | $MHz$ |
| $f_{IF}(F_b)$ | IF loss correction function | $dB$ |
| $\Gamma(r)$ | Radar Cross Section of reflections at a distance $r$ | $dBsm$ |
| $\Gamma_0$ | Radar Cross Section of the reference target | $dBsm$ |
| $\tilde{\Lambda}$ | Misalignment bias correction | $dB$ |
| $\lambda$ | Radar carrier wavelength | $m$ |
| $N$ | Number of iterations performed in a calibration experiment | |
| $P_r(r_0)$ | Power received from the target position $r_0$ | $dBm$ |
| $P_r(r)$ | Power received from distance $r$ | $dBm$ |
| $p_t$ | Radar transmitted power | $mW$ |
| $r$ | Distance from the radar | $m$ |
| $r_0$ | Distance between radar and reference target | $m$ |
| $F_0$ | IF frequency associated with the target distance | $m$ |
| $\sigma_A$ | Calibration uncertainty introduced by antenna properties | $dB$ |
| $\sigma_\epsilon$ | Standard deviation between all $C_{\Gamma i}^0$ values, used in the estimation of $\tilde{\Lambda}$ | $dB$ |

| | | |
|---|---|---|
| $\sigma_{\Gamma_0}$ | Uncertainty of the reference target RCS | $dB$ |
| $\sigma_i$ | Uncertainty in the estimation of each $C_{\Gamma i}^0$ value | $dB$ |
| $\sigma_{IF}$ | Uncertainty of the IF loss correction function | $dB$ |
| $\sigma_\Lambda$ | Uncertainty of the misalignment bias correction | $dB$ |
| $\sigma_{SCR}$ | Uncertainty introduced by clutter at the target position | $dB$ |
| $\sigma_T$ | Uncertainty of the temperature correction function | $dB$ |
| $\theta$ | Antenna beamwidth | $rad$ |
| $Z_e$ | Radar Equivalent Reflectivity | $dBZ$ |

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
