# Peer review of "Absolute Calibration method for FMCW Cloud Radars based on corner reflectors"

_Atmospheric Measurement Techniques, 2019_

## Short Comment (SC1) · 10 Mar 2020

**Comments on the manuscript "Absolute calibration method for FMCW cloud radars" by Toledo et al. 2020**

I find the manuscript to be devoted to the very important topic valuable for the cloud research community. The authors have done a good progress in establishing a calibration procedure aiming to reduce instrumentation uncertainty in cloud radar observations. I have several comments, which, I believe, will help the authors to improve the manuscript.

1. I strongly recommend to carefully check the units throughout the manuscript. Often the units do not fit. These are just occasions I noticed, there could be other.

    a. In the line 61 it is mentioned that Cgamma(T) is in dB. When I look into the Eq. 1a where this parameter is calculated I see a dimensionless number in the numerator and m^2*W in the denominator. 10log gives dB only when the ratio is dimensionless.

    b. Line 77: what does unreferenced dB mean? Power shall be in W, mW, dBm or something similar. It is not a unitless ratio and thus cannot be in just dB.

    c. Lines: 195 – 200: Power is in dB. Please see the comment 1b.

    d. Lines: 237 – 241: Power is in dB. Please see the comment 1b.

    e. Figures 3,4, power is in dB. Please see the comment 1b.

    f. RCS has different units throughout the manuscript. In line 113 it is in m^2, line 269 dBsm (is it dB related to square meter?), in Fig. 6 it is in dB (again cannot be because it is not unitless ratio).

2. Line 113: I would expect that the calibration targets are significant and actually the main source of uncertainties, which is not discussed in the manuscript. RCS are given up to the 4$^{th}$ significant digit. What are the uncertainties in these values? Were there any measurements of the targets made in an anechoic chamber? From a book about the radar reflectors, I know that the manufacturing precision is super critical for the RCS of corner reflectors. In the case of reflectors, which are much larger than the wavelengths, one degree deviation from perpendicularity of the reflector planes causes a change in RCS of about 10 dB. Since the aim of the proposed calibration procedure is to reach the calibration in the order of a few 0.1 dB (such an error could be easily caused by a tiny inperpendicularity), it would be very helpful to know how the authors evaluated this and what the uncertainty is.

3. Line 144: I would also recommend to mention which FFT window have been used for ranging, since it defines how many range bins to sum.

4. Since FMCW radars have intermediate frequency (IF) filters, the calibration can differ for different range bins due to different filter gains at different IF frequencies. How the frequency response of the IF filter is taken into account in the calibration method?

5. Correct me if I am wrong but as far as I understand $C^0_\gamma$ in lines 153-155 is different from the one calculated in Eq 4. In these lines $C^0_\gamma$ is calculated from each sample within an iteration, while the one in Eq. 4 is calculated from averages of several iterations.

6. Line 209: As far as I understand the lab pattern (Fig. 3b) characterizes only one antenna. In Fig. 3b I see a very high variability of the measurements (green and yellow dots). I do not see how based on these result the conclusion that the two antennas are parallel can be made.

7. Line 265: when I look into fig 5d I see that for some entire iterations the measurement points are not within 0.13 dB from the black line. Some iterations (like red one and violet one) have deviation exceeding 0.13 dB as well. Is the given uncertainty $\sigma_T$ = 0.13 dB reliable in this sense?

8. Are the numbers, given in line 311, resolution of the stepper or real angular accuracy? This can be tested if the same target angular position is reach from opposite rotation directions. Would the received power be the same?

9. Fig. 7a: the sharp edges on the left side of the blue curve and on the right side of the yellow curve look so much different from the rest of the curves in this figure. What causes such an effect?

10. Fig. 7a: The standard deviation of the yellow curve and the red one are close (0.09 and 0.13 respectively). But in the figure I see that the red curve is at least by a factor of 2 broader. Please double check the numbers.

11. From the table 2 I see that the uncertainty in the median values of the experiments A and B are < 0.03. 7a shows that the mean/median values for each iteration are distributed from -275.5 to -274.7. While 7b shows that all the iterations are in range of -274.2+/-0.2. If the number 0.03 is calculated for each iteration separately, which uncertainty component in the table 2 contains +/- 0.5 dB variability of the mean in the experiment A?

---

## Referee Comment (RC1) · Anonymous Referee #1 · 6 Apr 2020

The paper addresses a very important and difficult topic - calibration of cloud radars. While the principle of radar calibration might be simple, the practical realization and procedures are far from easy. I applaud the authors for undertaking this challenge. The paper describes experiments with corner reflectors fixed on a mast at the Sirta observatory in Palaiseau.

In its current form the paper is not fit for publication. Major revisions are needed:

There is lack of consistency in the use of units; for instance in equation 1a and 1b: - What is unreferenced dB? $P_t$ is in Watts, and $P_r$ in dB. - In eq. 1a, the temperature is explicit, but implicit in eq. 1b - La is in dB/km, but ro is not specified.

Since BASTA is a FMCW radar, the FFT in the determination of the range is crucial.

[Figure]

It introduces cross-talk, and usually time-windowing is applied to alleviate this at the expense of resolution. This needs more discussion in the paper.

To little is said about the properties of the calibration objects themselves. How well defined are they?

The temperature variation inside the radar is corrected for, but I do not think one can use the same temperature range for the calculation of Kˆ2. After all, the clouds are in an entirely different environment.

These fundamental aspects are missing in the paper. Since this can become a important reference paper for the European cloud radar community, I advise the authors to critically review it once more.

―――――――――――――――――――――

---

## Referee Comment (RC2) · Alexander Myagkov (Referee) · 12 Apr 2020

Please find my review in the attached pdf file.

Please also note the supplement to this comment:
https://www.atmos-meas-tech-discuss.net/amt-2019-498/amt-2019-498-RC2-supplement.pdf

---

## Author Comment (AC1) · 30 May 2020

Dear Dr. Alexander Myagkov,

Thank you for this careful and detailed review. Your comments were very valuable to improve the quality of the paper and the results. We provide here a point-by-point response to address each review comment (shown in blue color). Our responses appear in black color. In addition we explain all changes made to the manuscript. They can be seen in the manuscript document that includes change-tracking, attached to this file.

Additionally, we found that the reviewers comments were very helpful to improve the quality of our manuscript. Both reviewers are acknowledged in the acknowledgment section.

**1. Line 113: I would expect that the calibration targets are significant and actually the main source of uncertainties, which is not discussed in the manuscript. RCS are given up to the 4th significant digit. What are the uncertainties in these values? Were there any measurements of the targets made in an anechoic chamber? From a book about the radar reflectors, I know that the manufacturing precision is super critical for the RCS of corner reflectors. In the case of reflectors, which are much larger than the wavelengths, one degree deviation from perpendicularity of the reflector planes causes a change in RCS of about 10 dB. Since the aim of the proposed calibration procedure is to reach the calibration in the order of a few 0.1 dB (such an error could be easily caused by a tiny inperpendicularity), it would be very helpful to know how the authors evaluated this and what the uncertainty is. A reference with relevant information:**

- **Section 2.4 in Garthwaite et al 2015. The Design of Radar Corner Reflectors for the Australian Geophysical Observing System. (and references therein)**

The main objective of the paper is to present a calibration methodology. The methodology itself is not affected by the use of a real or theoretical target RCS. Actually, once the real target RCS is retrieved, any possible bias in the results can be corrected without changing the calibration method. We now state this more clearly in lines 190-196. The company that manufactures the targets declares having a cutting accuracy better than 0.1 mm and an alignment precision better than 0.1°, therefore we can expect a bias but it should be on the order of 1-2 dBsm.

We also include now how to account for the uncertainty of an eventual target characterization (eq. 6a and lines 231-234), and indicate that the uncertainty of the target calibration may increase the uncertainty in the results (lines 530-535).

Finally, as future work we now include the need of a target characterization in an anechoic chamber to correct any bias introduced by the use of the theoretical model (lines 598-600).

**2. I strongly recommend to carefully check the units throughout the manuscript. Often the units do not fit. These are just occasions I noticed, there could be other.**

a. In the line 61 it is mentioned that Cgamma(T) is in dB. When I look into the Eq. 1a where this parameter is calculated I see a dimensionless number in the numerator and m^2*W in the denominator. 10log gives dB only when the ratio is dimensionless.

b. Line 77: what does unreferenced dB mean? Power shall be in W, mW, dBm or something similar. It is not a unitless ratio and thus cannot be in just dB.

c. Lines: 195 – 200: Power is in dB. Please see the comment 1b.

d. Lines: 237 – 241: Power is in dB. Please see the comment 1b.

e. Figures 3, 4, power is in dB. Please see the comment 1b.

f. RCS has different units throughout the manuscript. In line 113 it is in m^2, line 269 dBsm (is it dB related to square meter?), in Fig. 6 it is in dB (again cannot be because it is not an unitless ratio).

The problem with the power units arises because power output in the BASTA radar is in an arbitrary power unit. We define this power unit as **dB(AU) = $10\log_{10}$(AU)**. The arbitrary unit defined as **AU** is proportional to watts multiplied by a unitless digital gain $k_d$, which depends on the digital signal processing configuration of the radar, such that **dB(AU) = dBW + $10\log_{10}(k_d)$**. Since the absolute calibration method will provide a calibration result that compensates this constant term, we did not work in transforming the power to standard physical units.

We now explained this detail in lines 72-76 . For consistency, now every power unit is defined in dB(AU) units, and therefore the RCS calibration is now in dB($AU^{-1}$ $m^{-2}$) and the reflectivity calibration is in dB( $mm^{-6}$ $m^{-5}$ $AU^{-1}$). This way, when the term is multiplied by reflected power and distance to the corresponding power, the result will be in the correct units (dBsm or dBZ).

All RCS values presented in the manuscript are now in dBsm units, both in text and figures. Line 84 also indicates that dBsm units are decibels referenced to a square meter. We also fixed a typo in Fig. 9 (prev fig 6). The maximum RCS indicated before in the label was of 28.28 dBsm, but it is actually 28.34 dBsm.

**3. Line 144: I would also recommend to mention which FFT window have been used for ranging, since it defines how many range bins to sum.**

During calibration we used a Hann time window, which is the default for the BASTA radar. This is now mentioned in line 178. Additionally, we include a new figure (Figure 3) to show which gates are used to estimate the target signal. The integration of additional gates increases the signal power by less than 0.01 dB, as indicated in lines 183-185.

**4. Since FMCW radars have intermediate frequency (IF) filters, the calibration can differ for different range bins due to different filter gains at different IF frequencies. How the frequency response of the IF filter is taken into account in the calibration method?**

Many thanks for this observation. Indeed we did not consider gain variations in the radar IF in the methodology. This comment motivated a significant improvement in the proposed Calibration Methodology, which now considers range variability in the receiver loss in addition to temperature effects. Because changes were important, they are introduced in several sections:

Abstract: included the IF correction in lines 8-9

Section 2: Equations used in Radar Calibration
This change is included in every mention of receiver losses as $L_r(T, r)$, and is therefore propagated to the RCS and reflectivity calibration terms as well, which now depend on temperature and range ($C_r(T, r)$, $C_z(T, r)$).
Lines 103-111 and equation (2) present how the IF correction is taken into account. Lines 114 to 119 explain in which order the calibration calculation is performed (first calibration at the mast position, then application of the IF correction).

Section 4: Methodology
Figure 2 now includes the IF correction as a block necessary for radar calibration. Lines 204-206 introduce how this correction function is retrieved.
Line 217 and eq. 6a now includes an IF correction uncertainty term.

Section 5.5: IF loss correction function
This is a new section explaining how we retrieve the IF correction function.

Section 6: Results
The results section now includes the value of the IF correction uncertainty and the IF correction functions retrieved for each calibration experiment in Table 2.

Section 7: Conclusions
The study of IF losses as a necessary part of calibration is now mentioned in lines 577-580.

**5. Correct me if I am wrong but as far as I understand C^0_gamma in lines 153-155 is different from the one calculated in Eq 4. In these lines C^0_gamma is calculated from each sample within an iteration, while the one in Eq. 4 is calculated from averages of several iterations.**

Yes, you understood correctly. To improve readability we modified the name of the $C_r^0$ of a single sample within an iteration to $C_{r_s}^0$ and added a short explanation, in lines 187-188.

**6. Line 209: As far as I understand the lab pattern (Fig. 3b) characterizes only one antenna. In Fig. 3b I see a very high variability of the measurements (green and yellow dots). I do not see how based on these result the conclusion that the two antennas are parallel can be made.**

The dots are the power received from the target normalized with respect to the maximum value and divided by 2, because we assume antennas are identical. This allows a comparison between data from scans and the laboratory antenna pattern. This is now explained in the caption of figure 4b (old 3b).

However, we did another revision of the scanning data and concluded that, at present, it is not possible to retrieve alignment information with an accuracy comparable to the antenna beam-width. This is now stated in lines 294-295. The reason is that the repeatability of the scanner positioning is not sufficient to allow a reliable retrieval under our current procedure.

Additionally, we now include a discussion on how parallax errors can influence the measurements (286-290), and indicate that calibration results are compatible with parallax errors smaller than the radar beamwidth (296-298). Since we don't have information on the exact alignment, we now mention the parallel antennas only as an hypothesis (245, 299-300).

Finally, we improved the calibration methodology by indicating how parallax errors can be taken into account, suggesting the addition of an additional range dependent correction function (300-301), and by introducing an uncertainty term representing the error in the antennas alignment estimation (eq. 6b, lines 243-245 and 301-302).

**7. Line 265: when I look into fig 5d I see that for some entire iterations the measurement points are not within 0.13 dB from the black line. Some iterations (like red one and violet one) have deviation exceeding 0.13 dB as well. Is the given uncertainty sigma_T = 0.13 dB reliable in this sense?**

To verify if data did follow a linear relationship, we did a new plot with the point density of all samples together. This figure has been added to the paper (Figure 7). In this figure it is easier to observe that deviated points are rather exceptional, with most points close to the regression. From this figure we think the 0.13 dB RMSE value is representative for most samples.

We also modified Figure 6 (D). Now it is only used to introduce the data set, with the linear fit shown in new Figure 7. This produced text changes in lines 351-355, and 360-371.

**8. Are the numbers, given in line 311, resolution of the stepper or real angular accuracy? This can be tested if the same target angular position is reach from opposite rotation directions. Would the received power be the same?**

This is the stepper resolution. Since the pointing algorithm relies on maximizing received power by scanning around the target, it is reasonable to assume that the uncertainty should be half of the stepper resolution. We agree that real angular accuracy is likely to be lower, as can be inferred from antenna scanning where received power variability is significant.

In any case, the RCS bias estimation of Figure 9 is just intended as an example to explain what is the misalignment bias and to show that its impact will most likely have a privileged direction.

The actual estimation of the bias correction is shown in section S3 of the supplementary material. This estimation relies on a sampling of all RCS distributions for randomly generated uncertainty sets. Specifically, for radar aiming we consider uncertainties from $0°$ to $0.375°$, which is three times the stepper resolution. We improved the explanation of this procedure in lines 460-472 and in the supplementary material.

**9. Fig. 7a: the sharp edges on the left side of the blue curve and on the right side of the yellow curve look so much different from the rest of the curves in this figure. What causes such an effect?**

This observation is very important. There was in fact a problem with the data used for the plot. When calculating the calibration results we selected the data from the contiguous hour with lowest variability, but the plot was incorrectly made using all the data from each target sampling iteration, as in the temperature characterization experiment.
We also took this correction as an opportunity to also improve the explanation of the methodology, adding lines 169-170 to indicate more precisely how data is chosen for the calibration estimation, and lines 200-203 to indicate that the temperature correction estimation is done using longer sampling times of the target signal.

The sharp lines corresponded to the beginning and end of the sampling period in cases where there was a systematic drift in the calibration constant value (for example a period under a fast temperature change in one direction).

The correction of this bug didn't affect the calibration values estimated for the 20 m mast, as they were calculated using the correct dataset, but did introduce a change of 0.01 dB in the 10 m mast retrievals. This is corrected in table 1 to guarantee consistency.

**10. Fig. 7a: The standard deviation of the yellow curve and the red one are close (0.09 and 0.13 respectively). But in the figure I see that the red curve is at least by a factor of 2 broader. Please double check the numbers.**

This was caused by the same problem explained in comment 9.

**11. From the table 2 I see that the uncertainty in the median values of the experiments A and B are < 0.03. 7a shows that the mean/median values for each iteration are distributed from -275.5 to -274.7. While 7b shows that all the iterations are in range of -274.2+/-0.2. If the number 0.03 is calculated for each iteration separately, which uncertainty component in the table 2 contains +/- 0.5 dB variability of the mean in the experiment A?**

Since the effective RCS distribution for our system alignment is not gaussian, we couldn't use the standard deviation between iterations as a direct estimator of uncertainty.

Rather, what we did was to perform an estimation of the effective RCS distribution for our system using the observed variability between calibration results as input information. This effective RCS distribution is used to compute the most likely calibration value consistent with our observed spread through the estimator of Eq. (5), and therefore the uncertainty of this procedure will be represented by the uncertainty in each term of this estimator.

These terms are the uncertainty propagated through the calculation of the average calibration result ($\Sigma\sigma_i/N^2$ and $\sigma_T/N$) plus the uncertainty in the bias correction term ($\sigma_\Lambda$). This is now indicated explicitly in lines 468-472.

**12. I think Lat in Eqs . 1b and 3b is also range dependent. Please indicate this.**

We inserted this correction in the requested equations. We also corrected this same error in equation (7). Equations (1b) and (3b) were re-organized to correct for the ambiguity introduced when using both r and r0 (explained in the next point).

**13. What is the difference between r0 and r in Eqs. 1b and 3b?**

There was indeed an ambiguous use of r and $r_0$. This is corrected. Now r always represents the distance from the radar. It is used, for example, to indicate that a term is a function of distance.
Along this change, $r_0$ is now only used to indicate the distance of the target (from the radar). It is used for example to indicate that a certain variable dependent on distance is evaluated at the target position. This is now written in lines 93-94.
The distinction between r and $r_0$ is also explained in the newly added table of symbols.

**14. Sometimes it is hard to follow the text because of a huge number of symbols. I therefore strongly recommend to add a table with a short description of all used symbols.**

Thanks for this recommendation. We now include a table of symbols at the end of the manuscript to improve text readability.

**15. In the Eq. 5 the authors assume that errors are not correlated from iteration to iteration. But if for example two consecutive iterations are made under similar conditions I would expect a certain correlation. In this case variances would not be reduced by factor of N^2 and N but by a smaller factor (Leith 1973, The standard error of time-average estimates of climatic means).**

While developing the mathematical method for the estimation of the calibration coefficient $C_r^0$ , we made the underlying hypothesis that variations in the estimation of this value between different iterations are exclusively explained by an underlying probability distribution of effective RCS values. This distribution is generated by the alignment uncertainty of the experimental setup.

The method also includes the hypothesis that the resulting effective RCS after each system realignment is not correlated with the result of the previous one. As indicated in point 11 and as is now indicated in lines 460-472, by following this hypothesis we arrive to the calibration estimator of Eq. (5). During the derivation of this estimator, we get that the uncertainty is distributed between the terms $\Sigma\sigma_i/N^2$, $\sigma_T/N$ and $\sigma_\Lambda$. The details of this derivation are in section S3 of the supplementary material.

To have meaningful results, we implemented many measures to have a system that behaves as close as possible to the theoretical model. Specifically, before each sampling iteration we purposely misalign the system very far from the operating conditions. This should reset the previous alignment state. Then, we follow the exact same alignment protocol each time, to ensure that the underlying distribution of the effective RCS remains unchanged for all iterations. This procedure is in fact a key aspect of the proposed calibration methodology.

[revised manuscript text omitted]

---

## Author Comment (AC2) · 30 May 2020

Dear Anonymous Referee,

Thank you for reviewing our document. Your comments helped us to significantly improve the article quality. We provide here a point-by-point response to address each of your comments, in green color. Our responses appear in black color. In addition we explain all changes made to the manuscript. They can be seen in the manuscript document that includes change-tracking, attached to this file.

We found that the reviewers comments were very helpful to improve the quality of our manuscript. Both reviewers are acknowledged in the acknowledgment section.

**1. There is lack of consistency in the use of units; for instance in equation 1a and 1b:**
- **What is unreferenced dB? Pt is in Watts, and Pr in dB. - In eq. 1a, the temperature is explicit, but implicit in eq. 1b**
- **La is in dB/km, but ro is not specified.**

Thank you for this comment, which helps us to improve the readability and consistency of our manuscript. To improve our article we made several changes, explained below.

- We changed the variable $P_t$ in caps for $p_t$ to emphasize that this term is in a linear power scale, with respect to the received power $P_r(r)$ which is in logarithmic scale. The terms are also described in the newly added table of symbols.

- The unreferenced dB problem arises because the power output in the BASTA radar is in an arbitrary power unit. We define this power unit as **dB(AU) = 10log$_{10}$(AU)**. The arbitrary unit defined as **AU** is proportional to watts multiplied by a unitless digital gain $\mathbf{k_d}$, which depends on the digital signal processing configuration of the radar, such that **dB(AU) = dBW + 10log$_{10}$(k$_d$)**. Since the absolute calibration method will provide a calibration result that compensates this constant term, we did not work in transforming the power to standard physical units.
Rather, what we did is to explain this detail about the units in lines 72-76. Then, for consistency, we propagated this change to every power unit displayed in the manuscript, which are now defined in dB(AU) units.

With these changes the units of all equations match, notably the RCS calibration is now in dB(AU$^{-1}$ m$^{-2}$) and the reflectivity calibration is in dB( mm$^{-6}$ m$^{-5}$ AU$^{-1}$) units. This way, when the term is multiplied by reflected power and distance to the corresponding power, the result will be in the correct physical units (dBsm or dBZ).

- First it is important to mention that a new range dependency is introduced in the calibration constant, in addition to the temperature dependency. This range dependency is included to compensate gain effects introduced by non-ideal amplifiers in the Intermediate Frequency (IF) circuit of the receiver. Both the temperature and the IF effects will modify the system gain, and therefore the measured power. The temperature and range dependency is introduced in the calibration constant to explicitly indicate that these effects are corrected in

raw power samples. This explanation is now written in lines 90-92, just after introducing Eq. 1b.

- $L_a$ units were incorrect. It represents a pure loss and therefore it is now in dB units. We also now explicit the range dependency of this term in all equations: $L_a = L_a(r)$.

- $r_0$ is the distance between the radar and the reference reflector. This definition was not respected in many parts of the manuscript so we performed a thorough revision to make sure its use remains consistent. We also added an explanation on the meaning of $r_0$ in lines 93-94 and in the table of symbols.

**2. Since BASTA is a FMCW radar, the FFT in the determination of the range is crucial. It introduces cross-talk, and usually time-windowing is applied to alleviate this at the expense of resolution. This needs more discussion in the paper.**

To improve our description of the measurement conditions we now state that we used a Hann time window during the calibration, in line 178. Additionally, to improve the description on how we perform the experiment, we include a new figure (Figure 3). In this figure we show which gates are used to estimate the target signal. There it is possible to see that cross-talk effects become negligible approximately 50 meters away from the radar, and that the target signal is easily discernible from nearby objects (trees in Figure 3 b). We also added the references of *Richardson 1978* and *Doviak, R. J. and Zrnic 2006* which can be useful to learn more about time windowing and the discrete fourier transform.

**3. Too little is said about the properties of the calibration objects themselves. How well defined are they?**

In our paper we use a theoretical model of RCS. This is because the main objective of the paper is to present a calibration methodology. The use of this theoretical model may introduce a bias in the results, but it can be corrected once a target characterization is performed. We now state this more clearly in lines 190-196.

The company that manufactures the targets claims to have a cutting accuracy better than 0.1 mm and an alignment precision better than 0.1°, therefore we can expect a bias but it should be in the order of 1-2 dBsm. We also include a reference on what is the impact of misalignment in the target RCS (*Doerry et al. 2009*).

Other additions to the methodology are how to account for the uncertainty of using a characterized target RCS instead of the theoretical model in eq. 6a and lines 231-234 and 530-535.

Finally, as future work we now include the need of a target characterization in an anechoic chamber to correct any bias introduced by the use of the theoretical model (lines 598-600).

**4. The temperature variation inside the radar is corrected for, but I do not think one can use the same temperature range for the calculation of Kˆ2. After all, the clouds are in an entirely different environment.**

It is true that the dielectric factor K depends on the properties and temperature of the back-scatterers. In fact what we do is to calculate a reference calibration term for K at 5°C which has to be corrected when performing retrievals. The application of this correction in the calibration constant is simple (it is enough to replace the K value in the calculation), but requires the estimation of cloud properties. We improved this explanation in lines 236-242 and added three references which explain how K can be estimated in retrievals (*Sassen 1987; Liebe et al. 1989; Gaussiat et al. 2003*).

**5. These fundamental aspects are missing in the paper. Since this can become a important reference paper for the European cloud radar community, I advise the authors to critically review it once more.**

Thank you for reviewing our paper, and for the encouragement. We did a complete revision of our manuscript and this resulted in many improvements to the methodology and result analysis. We believe that the methodology we propose now is more comprehensive, since it considers more uncertainty sources and indicates how the addition of missing information at the writing time can be implemented, to reduce the possible bias arising from some assumptions.

[revised manuscript text omitted]

---

## Referee Report (RR1)

Dear authors,

Thanks a lot for revising the manuscript. I still have major concerns, which have to be solved before the manuscript can be published. After modifications, I would like to look through the revised version.

Below I address authors' comments. The authors responses are in blue, my new comments/replies are in black.

1. The main objective of the paper is to present a calibration methodology. The methodology itself is not affected by the use of a real or theoretical target RCS. Actually, once the real target RCS is retrieved, any possible bias in the results can be corrected without changing the calibration method. We now state this more clearly in lines 190-196. The company that manufactures the targets declares having a cutting accuracy better than 0.1 mm and an alignment precision better than 0.1°, therefore we can expect a bias but it should be on the order of 1-2 dBsm. We also include now how to account for the uncertainty of an eventual target characterization (eq. 6a and lines 231-234), and indicate that the uncertainty of the target calibration may increase the uncertainty in the results (lines 530-535). Finally, as future work we now include the need of a target characterization in an anechoic chamber to correct any bias introduced by the use of the theoretical model (lines 598-600).

Citation: "However, since at the writing time we do not have an experimental characterization for our targets, we rely on the: theoretical model. This is not a major issue because, once an experimental characterization of the target becomes available, it can be used to correct any calibration bias by rectifying the value of Gamma used in the calculations"

I do not agree with the authors. What is described in the manuscript is a method (i.e. description of steps to get knowledge), not a methodology (analysis of a set of methods). And in my opinion a calibration method is worth nothing without a proper characterization of a calibration target. I think this is the first thing one should do for the radar calibration – characterize the reference target. Currently it sounds to me, that after the proposed calibration procedure another calibration steps would be required (characterization of the target and application of another bias correction) when the target is measured in a chamber. The authors claim, "A detailed analysis enabled the design of a calibration methodology which can reach a cloud radar calibration uncertainty of **0.3 dB based on the equipment used in the experiment**". This can be misleading for a reader. The authors do not reach the claimed value (0.3 dB) in the current work. As authors estimate, the real uncertainty is not known at the moment and may be in the order of 2 dB (dBsm are not proper units here, since this value is unitless in linear scale). I suggest two ways to solve this problem:
- Authors characterize the target in a chamber and add these results (cross section and its uncertainties) in the manuscript.
- Authors use sigma_rcs = 2 dB in Eq. 6a, reevaluate the results, and write explicitly in the abstract, main text, and conclusions that the uncertainty of the proposed method **at the current stage** is not better than ... dB due to uncharacterized reference target. Otherwise, it is not honest to neglect a large uncertainty source just because it is not characterized.

I would strongly recommend the authors to follow one of these ways.

2. The problem with the power units arises because power output in the BASTA radar is in an arbitrary power unit. We define this power unit as $dB(AU) = 10\log_{10}(AU)$. The arbitrary unit defined as $AU$ is proportional to watts multiplied by a unitless digital gain $k_d$, which depends on

the digital signal processing configuration of the radar, such that $\mathtt{dB(AU) = dBW + 10 log_{10}(k_d)}$ . Since the absolute calibration method will provide a calibration result that compensates this constant term, we did not work in transforming the power to standard physical units. We now explained this detail in lines 72-76 . For consistency, now every power unit is defined in dB(AU) units, and therefore the RCS calibration is now in dB(AU $_{-1}$ m $_{-2}$ ) and the reflectivity calibration is in dB( mm $_{-6}$ m $_{-5}$ AU $_{-1}$ ). This way, when the term is multiplied by reflected power and distance to the corresponding power, the result will be in the correct units (dBsm or dBZ). All RCS values presented in the manuscript are now in dBsm units, both in text and figures. Line 84 also indicates that dBsm units are decibels referenced to a square meter. We also fixed a typo in Fig. 9 (prev fig 6). The maximum RCS indicated before in the label was of 28.28 dBsm, but it is actually 28.34 dBsm.

The introduced changes are even more confusing. The calibration terms characterize a ratio of a real measure over the calculated one. Therefore, calibration terms must be unitless in linear scale and in dB in the logarithmic scale. I do not understand what a calibration term in dB(AU^-1m^-2) means.

The lines 72 – 76 are confusing. It is stated that kd is included to account for the units of the measured power which is in 10*log10(AU). One sentence later it is stated that kd is unitless. If it is unitless then the equation 1a has problems with units again. The nominator is unitless, the denominator has units of m^2*W as it was in the original version. I kindly ask the authors to carefully reconsider the units again.

In fact, the previous version was better, the only problem was with units notation, i.e. dB was used instead of dBm and dBsm (please see my previous comments). Please modify the units in such a way that the calibration factors are given in dB (unitless in linear scale). And please modify the units throughout the manuscript accordingly.

3. During calibration we used a Hann time window, which is the default for the BASTA radar. This is now mentioned in line 178. Additionally, we include a new figure (Figure 3) to show which gates are used to estimate the target signal. The integration of additional gates increases the signal power by less than 0.01 dB, as indicated in lines 183-185.

Thanks. It is clear now.

4. This change is included in every mention of receiver losses as L r (T, r), and is therefore propagated to the RCS and reflectivity calibration terms as well, which now depend on temperature and range (C Γ (T, r), C Z (T, r)).

I would recommend to use IF instead of r because for a different chirp configuration (slope) the relations between IF bins and range gates may change.

In this newly added section the authors, as far as I understand, assume that during the 'passive' observations the power variability along IF depends only on gain changes. In general case this is not true:

$$Pr(IF) \sim G(IF)*(Tsys(IF) + Tamb)$$

Here Pr(IF) is the received power at IF in W, G(IF) is the linear gain of the receiver chain at IF (unitless), Tsys(IF) is system noise temperature at IF in K, Tamb is brightness temperature of the sky (or an object the radar was pointed to) in K, ~ is the proportionality sign. From this equation one can see that the received power depends on two parameters, namely the gain and the system noise temperature. If I understand right, the authors did so-called single point calibration. Using the single point calibration is it

not possible to separate the gain and the system noise temperature. Therefore, typically two-point calibrations are used in radars and radiometers. Also the authors need to know Tamb (at least with respect to Tsys(IF)). I kindly ask the authors to clarify how they took these aspects into account to calibrate all the IF bins of the radar receiver.

6. However, we did another revision of the scanning data and concluded that, at present, it is not possible to retrieve alignment information with an accuracy comparable to the antenna beam-width. This is now stated in lines 294-295. The reason is that the repeatability of the scanner positioning is not sufficient to allow a reliable retrieval under our current procedure. Additionally, we now include a discussion on how parallax errors can influence the measurements (286-290), and indicate that calibration results are compatible with parallax errors smaller than the radar beamwidth (296-298). Since we don't have information on the exact alignment, we now mention the parallel antennas only as an hypothesis (245, 299-300). Finally, we improved the calibration methodology by indicating how parallax errors can be taken into account, suggesting the addition of an additional range dependent correction function (300-301), and by introducing an uncertainty term representing the error in the antennas alignment estimation (eq. 6b, lines 243-245 and 301-302).

The assumption on parallel antennas can lead to large uncertainties. The problem with two antennas is that it is possible to measure the pattern of the receiving antenna with an external transmitter but it is often not possible to measure the transmitting antenna. Basically, with the proposed method only two points of the possible range dependent bias are characterized.

Instead of leaving this large uncertainty source untouched, I would encourage the authors to make a relatively simple estimation of possible impacts (just theoretical calculations, taking into account different divergences (magnitude and direction) of the two antennas and bias measurements at two distances). This would definitely improve the quality of the manuscript. The result of this theoretical estimation would give a proxy for sigma_a in Eq. 6a which is currently, if I understand it right, completely neglected.

Just to better understanding I give a couple of figures:

[Figure]

On the left figure you can see different divergence directions. On the right figure I illustrate the impact (qualitatively). The authors could perform such calculations and give an estimate for sigma_a (maximum divergence from 0 dB line).

7. To verify if data did follow a linear relationship, we did a new plot with the point density of all samples together. This figure has been added to the paper (Figure 7). In this figure it is easier to observe that deviated points are rather exceptional, with most points close to the regression. From this figure we think the 0.13 dB RMSE value is representative for most samples. We also modified Figure 6 (D). Now it is only used to introduce the data set, with the linear fit shown in new Figure 7. This produced text changes in lines 351-355, and 360-371.

In Fig. 7 the authors just masked the problem I am talking about. I agree that a majority of samples follow the linear model. But some complete iterations (like green points in Fig 6d) are off by more than 0.5 dB.

---

## Referee Report (RR2)

Dear authors, thanks a lot for considering my comments. The manuscript has been significantly improved. However, I have a major concern regarding formulas the authors give. In addition, I have some minor comments. Please find my comments below. The author's latest comments are in blue color. My new comments are in green.

**Author's response:** We agree about the need of characterizing the target to provide final uncertainty results. However, we do not agree that the lack of the target characterization cancels the validity of the results, since we used a theoretical model of target RCS that has all the properties that a calibrated target would have, except the absolute values.

The results presented in the article enable the identification of several uncertainty sources, as well as their relative contribution to the experiment uncertainty. This information can be very valuable to design future calibration experiments based on reference reflectors, whether they are mounted on masts or held by other means, such as UAVs. The underlying principles remain the same.

We also add that our objective is not to claim we have a reference instrument, but to present all the information and advancements obtained from our experimental campaigns, specially in uncertainty characterization. For example, with the results we can quantitatively compare two different experimental setups, finding different factors limiting uncertainty for each (SCR for the 10 m mast, alignment for the 20 m mast). Additionally, we are not aware of any other published methodology of radar calibration that considers the bias introduced due to misalignment between target and radar. For us this work is a step towards more precise calibration methodologies, and we expect it to act as a reference to improve the preparation of future calibration experiments. Because it is true that at this stage we can only do a rough estimation of RCS uncertainty, we agree to highlight this explicitly. This is now stated/included in:

Line 14 of the Abstract.
Lines 197-200
Line 229
Lines 507-509
Calibration result for all experiments (lines 512-531)
Table 2
Lines 585-587
This also implied modifications to some text in the article to remain consistent:
Lines 18-20 of the abstract.
Lines 54-55 of the Introduction
Lines 153-154
We also added an estimation of the maximum uncertainty in RCS characterization required to reach a calibration uncertainty of 0.5 dB in lines 592-597.

**Comment:** Thanks for these changes. The modifications you have introduced make it much clearer for a reader that the main source of uncertainties is the characterization of the used reflectors and that this is not covered in this study. Just a few minor comments here:
- Please be consistent with the goal value. It is 1 dB in the abstract (line 20) but 0.5 dB throughout the text (e.g. lines 55, 593).
- In lines 593 – 594 you apply the formula of the std of a sum of two uncorrelated variables, if I understand it right (sqrt(0.4^2+0.3^2) = 0.5). I would agree with it if one would use a newly manufactured reflector every time, but most likely, it will be just a single one so the bias due to the corner reflector will be constant (zero variance, the formula is not applicable). In this case,

the reflector contributes to the systematic error, while the effects considered in the manuscript characterize the random error. I would recommend making this clear.

- Since the study makes conclusions on the total uncertainties with a number of assumptions (i.e. parallel antennas, flat noise figure, reflector with known cross section), please consider a change of the title of the manuscript to something like "Aspects of cloud radar calibration based on corner reflectors".

**Author's response:** Following this suggestion, we now present the received power units to dBm and the calibration terms to dB. This led to the recalculation of the calibration terms absolute value.
Modifications:
- Because of this improvement, lines 79-80 and 82-84 explaining these arbitrary power units are no longer necessary and were removed.
- Eq. (2a) and line 542 were modified to remove the unnecessary digital gain term kd.- Writing of the units corrected in lines 97, 112, 127, 142 and the Glossary.
- Absolute value of power/calibration terms corrected in lines 261, 264, 265, 267, 319-321, Table 1, and lines 512-531 of the calibration results.
- Additionally, Figures 3, 4(a), 5, 6 and 10 were modified to remain consistent with the power units.

**Comment:** Here I still have a major concern. I would like to thank the authors for their efforts, but the equation 2 is still wrong. It cannot give dB units, because the ratio in the parenthesis is still not unitless. Please note, that this is already the third time I ask to adjust all the terms properly.

I try to do my best to explain what I expect if I were using the method proposed by the authors. Typing long formulas in Word is a bit inconvenient. Therefore, I wrote my considerations on paper.

$P_r = P_m \cdot C_r$ (Eq. 1)

$P_r$ – real (expected from the radar equation) received power [W]

$P_m$ – measured received power [W]

$C_r$ – calibration term [unitless]

In log scale:

$P_r \, [dBm] = P_m \, [dBm] + C_r \, [dB]$ (Eq. 2)

In the case of an ideal radar we expect:

$$P_r = \frac{\pi^3}{16 \log_e 2} \cdot \frac{P_t \cdot h}{\lambda^2} \cdot G_t \cdot G_r \cdot \theta^2 \frac{1}{R^2} |K|^2 Z \quad (Eq. 3)$$

Eq. 3 is for meteorological targets and is taken from Probert–Jones 1964.

$P_t$ – transmitted power [W]

$h$ – range resolution [m]

$\lambda$ – wavelength [m]

$G$ – antenna gain [unitless]

$R$ – range [m]

$|K|^2$ – dielectric factor [unitless]

$Z$ – reflectivity [mm$^6$m$^{-3}$]

not estimated

$$P_m = P_r \cdot \boxed{\frac{L_t}{L_r} \cdot \frac{G_t^{real}}{G_t^{assum.}} \cdot \frac{G_r^{real}}{G_r^{assum}} \cdot \frac{\theta_{real}^2}{\theta_{assum.}^2} \cdot \frac{1}{L_o}} \boxed{\frac{1}{L_{||}} \cdot \frac{L_{at}^{real}}{L_{at}^{assum.}} \cdot \frac{Z^{real}}{Z^{assum.}}} \quad (Eq. 4)$$

estimated in the study          $C_r$ (unitless)

$L_t$ – losses in transmitter path

$L_r$ – losses in receiver path

$L_o$ – losses due to the antenna overlap

$L_{||}$ – losses in the case antennas are not parallel

In the case of a point target $\frac{Z^{real}}{Z^{assum}}$ should be $\frac{\Gamma^{real}}{\Gamma^{assum.}}$

where $\Gamma$ is the reflector cross-section in m$^2$

Indecks 'real' and 'assum' correspond to real (true) values and assumed ones, respectively.

I forgot to write one term. Lat is the attenuation due to propagation to the target and back.

In my point of view, the authors should use the calibration term Cgamma as in Eq. 4 in the drawing. It is clear how to use it in order to correct the measurements (it works in the same way for power and reflectivity values) and it is unitless as it should be. I marked two components of the calibration term, the one characterized in the manuscript (greed rectangle) and the one with certain assumptions (red rectangle). I believe such a separation in the very beginning of the manuscript would help a reader to understand which effects are considered and which are not.

**Author's response:** We originally used range because BASTA-Mini has only 4 standard operational modes, so it was not a very important distinction. Nevertheless, we agree that this formulation would improve the generality of the method, and consequently we modified all range dependent terms by terms depending on the IF frequency $F_b$.
Besides, we added a short explanation of the $F_b$ term in lines 75-81 and the new Equation (1), which indicates how $F_b$ it is associated with the range r.

**Comment:** Thanks, it is better now. The method in general can be used for other FMCW radars as well and a reader should understand that the receiver loss depends on IF and not on range.

**Author's response:** In this section we do not intend to estimate the absolute value of gain at the IF, but rather to quantify relative gain changes with respect to the calibrated IF frequency (associated with the target position). To do this using passive observations, we had to make the assumption of a constant noise power, both from the system and from environment in the 12 MHz bandwidth of the receiver. This assumption is reasonable because components used in the receiver have a much larger bandwidth (for example for the Low Noise Amplifier (LNA) it is of 35 GHz).
We indicated briefly this assumption in the previous version, but we didn't mention the impact it could have on uncertainty. This is now estimated from the LNA specifications. Ist variability of gain and noise figure in the 12 MHz bandwidth used is smaller than 0.1 dB. Since LNA are typically the main source of system noise in the receiver, we consider that 0.1 dB is a safe estimation of the uncertainty introduced by assuming a constant system noise in the IF bandwidth. This term dominates the RMSE between the fit and data, and the inter-period variability, thus we now define the IF correction function uncertainty to be of 0.1 dB.
Despite this, we agree that two point calibrations are highly desirable because they enable the retrieval of receiver absolute gain and system noise. This is now indicated in the text. Thus, article changes are:
- More accurate explanation of passive observations (lines 365-368, equation 9a)
- Brief indication of the benefits of performing two point calibrations for receivers (lines 370-373)
- Explanation of the constant system noise assumption and introduced uncertainty (lines 374-383, equation 9b)
- Clarification in the explanation on how the IF correction function is retrieved (lines 384-396, line 406-408).
- Final uncertainty of the IF correction function (lines 413-414)

**Comment:** A couple of major concerns:
- Lines 265-267: formulation is wrong. The very first component in the receiver chain is a low noise **amplifier**. The receiver amplifies the received signal not reduces the power level. Despite on the amplification LNA reduces the signal-to-noise ratio and this is characterized by its noise figure. I recommend to use the formula I gave last time for the proper explanation.

- Formula 9: If noise powers are in linear units, why dB values of losses are subtracted? If noise powers are in dBm they cannot be summed.

Also a minor concern:

- Line 376: please be careful here. I completely agree that noise figure of LNA can be assumed flat. Since noise figure of mixers, active IF filters and ADC units can be very high, these components can still contribute to the noise figure despite the amplification of LNA. The question is how about other components and standing waves, which can produce wavy shape (> 100 K variability) of the system noise temperature even within several MHz bandwidth? If this was considered please mention, if not please explain that other effects caused by other components of the receiving chain or standing waves can affect the assumption of the constant noise temperature.

- Taking into account the three comments, I recommend a revisiting the section 5.5 lines 364 - 384.

**Author's response:** Thank you for this proposal, we believe it is a very good idea with good potential. Because of this, we performed several theoretical calculations to check if we could estimate a range of possible antenna misalignment angles and the associated uncertainty with our data. Summarizing, our results show that the experimental setup used is not appropriate for this measurement, but they also indicate us a path to perform such experiments in the future. Since the targets used at 196 and 376.5 are different, the uncertainty in the **calibration coefficient difference** at both distances is very large (~3 dB). This uncertainty makes it impossible to bound the possible alignment within 1.5 degrees, which is our antenna characterization width. This large uncertainty comes mostly from the use of two different calibration targets. This decision was made because the experiment was designed to applicability of the absolute calibration method for different experimental setups, and because the proposed experiment was not considered at the time. Given that the proposed experiment was not done during the campaigns presented in the article, we have no way to gather any additional information on antenna alignment for that period. Thus, we leave this section unchanged with respect to the previous version. Yet, with the theoretical calculations we found that if we get an uncertainty in the order of 0.5 dB when comparing calibration constants at these two different distances, antenna misalignment could be constrained to values ranging in the order of tenths of degree. This could be achieved, for example, by using the 20 meter mast setup at both distances using the same reflector each time. It is worth noting that the tools developed for this analysis now enable the design of an experiment with optimized parameters for this retrieval. Taking all this into consideration, in our opinion the potential of the proposed experiment indicate that it must be further studied for its implementation in future calibration campaigns.

**Comment:** I understand the point of the authors. I would recommend to do the following. Please summarize the effects which you have characterized in your study and write that the uncertainty you have found are only related to these effects. Also specify explicitly a list of effects to be characterized in the future (target, IF dependency of noise figure, parallelism of antennas, etc).

**Author's response:** This happens because, to capture the widest range of possible temperatures, we had to use longer time series of data. The use of longer time series introduced some points measured under suboptimal conditions. For example, we have observed that high wind speeds lead to larger variabilities in the calibration value due to oscillations of both the mast and the radar. Meanwhile, drizzle adds a time dependent bias, most likely caused by changes in wet radome (and wet target) attenuation over time. An effort was done to clean the dataset, but inevitably some noisy data points

remained. This is now stated more clearly in lines 345 to 348. Therefore, to estimate the temperature correction function and its uncertainty we did an statistical analysis of data, and then used the RMSE between the model and data points as the estimation of model uncertainty. To bound the uncertainty value we calculated the RMSE of the model for each degree of deviation from the reference temperature, obtaining a RMSE range between 0.07 to 0.23 for 0 and +3 degrees of deviation respectively. We also checked the bias per degree for each iteration and for all the dataset, and found out that its mean value is always within +- 0.2 dB with respect to the model. Therefore, we now state that the temperature correction function uncertainty is less or equal to 0.23. This change is reflected in lines 353-357. This is also true for the mentioned case, were most points are covered by the other iterations data. The larger spread reaching deviations of 0.5 dB in this case are caused by short period of drizzle that happened in this iteration. However, since this data, and the rest of data that deviates from the model is also included in the calculation of RMSE, we think this is a reliable criteria for the estimation of the temperature correction function uncertainty.

**Comment:** Thanks for the explanation. It is clear now.

---

## Author Response (AR2)

Dear Dr. Alexander Myagkov,

Thank you for your new feedback. It was very useful to improve the article quality and clarity, improving the reliability, and thus the value, of our results. As before, we explain changes done to the manuscript here and attach a manuscript document with change-tracking.

Our previous responses are in blue, your new comments are in brown and our new responses are in black.

**1.**The main objective of the paper is to present a calibration methodology. The methodology itself is not affected by the use of a real or theoretical target RCS. Actually, once the real target RCS is retrieved, any possible bias in the results can be corrected without changing the calibration method. We now state this more clearly in lines 190-196. The company that manufactures the targets declares having a cutting accuracy better than 0.1 mm and an alignment precision better than 0.1°, therefore we can expect a bias but it should be on the order of 1-2 dBsm. We also include now how to account for the uncertainty of an eventual target characterization (eq. 6a and lines 231-234), and indicate that the uncertainty of the target calibration may increase the uncertainty in the results (lines 530-535). Finally, as future work we now include the need of a target characterization in an anechoic chamber to correct any bias introduced by the use of the theoretical model (lines 598-600).

Citation: "However, since at the writing time we do not have an experimental characterization for our targets, we rely on the: theoretical model. This is not a major issue because, once an experimental characterization of the target becomes available, it can be used to correct any calibration bias by rectifying the value of Gamma used in the calculations"

I do not agree with the authors. What is described in the manuscript is a method (i.e. description of steps to get knowledge), not a methodology (analysis of a set of methods). And in my opinion a calibration method is worth nothing without a proper characterization of a calibration target. I think this is the first thing one should do for the radar calibration – characterize the reference target. Currently it sounds to me, that after the proposed calibration procedure another calibration steps would be required (characterization of the target and application of another bias correction) when the target is measured in a chamber. The authors claim, "A detailed analysis enabled the design of a calibration methodology which can reach a cloud radar calibration uncertainty of 0.3 dB based on the equipment used in the experiment". This can be misleading for a reader. The authors do not reach the claimed value (0.3 dB) in the current work. As authors estimate, the real uncertainty is not known at the moment and may be in the order of 2 dB (dBsm are not proper units here, since this value is unitless in linear scale). I suggest two ways to solve this problem:
- Authors characterize the target in a chamber and add these results (cross section and its uncertainties) in the manuscript.
- Authors use sigma_rcs = 2 dB in Eq. 6a, reevaluate the results, and write explicitly in the abstract, main text, and conclusions that the uncertainty of the proposed method at the current stage is not better than ... dB due to uncharacterized reference target. Otherwise, it

is not honest to neglect a large uncertainty source just because it is not characterized. I would strongly recommend the authors to follow one of these ways.

We agree about the need of characterizing the target to provide final uncertainty results. However, we do not agree that the lack of the target characterization cancels the validity of the results, since we used a theoretical model of target RCS that has all the properties that a calibrated target would have, except the absolute values.

The results presented in the article enable the identification of several uncertainty sources, as well as their relative contribution to the experiment uncertainty. This information can be very valuable to design future calibration experiments based on reference reflectors, whether they are mounted on masts or held by other means, such as UAVs. The underlying principles remain the same.

We also add that our objective is not to claim we have a reference instrument, but to present all the information and advancements obtained from our experimental campaigns, specially in uncertainty characterization. For example, with the results we can quantitatively compare two different experimental setups, finding different factors limiting uncertainty for each (SCR for the 10 m mast, alignment for the 20 m mast). Additionally, we are not aware of any other published methodology of radar calibration that considers the bias introduced due to misalignment between target and radar. For us this work is a step towards more precise calibration methodologies, and we expect it to act as a reference to improve the preparation of future calibration experiments.

Because it is true that at this stage we can only do a rough estimation of RCS uncertainty, we agree to highlight this explicitly. This is now stated/included in:
Line 14 of the Abstract.
Lines 197-200
Line 229
Lines 507-509
Calibration result for all experiments (lines 512-531)
Table 2
Lines 585-587

This also implied modifications to some text in the article to remain consistent:
Lines 18-20 of the abstract.
Lines 54-55 of the Introduction
Lines 153-154

We also added an estimation of the maximum uncertainty in RCS characterization required to reach a calibration uncertainty of 0.5 dB in lines 592-597.

**2.** The problem with the power units arises because power output in the BASTA radar is in an
arbitrary power unit. We define this power unit as dB(AU) = 10log 10 (AU) . The arbitrary unit

defined as AU is proportional to watts multiplied by a unitless digital gain $k_d$, which depends on the digital signal processing configuration of the radar, such that dB(AU) = dBW +10log$_{10}$($k_d$). Since the absolute calibration method will provide a calibration result that compensates this constant term, we did not work in transforming the power to standard physical units. We now explained this detail in lines 72-76 . For consistency, now every power unit is defined in dB(AU) units, and therefore the RCS calibration is now in dB(AU$^{-1}$ m$^{-2}$) and the reflectivity calibration is in dB( mm$^{-6}$ m$^{-5}$ AU$^{-1}$ ). This way, when the term is multiplied by reflected power and distance to the corresponding power, the result will be in the correct units (dBsm or dBZ). All RCS values presented in the manuscript are now in dBsm units, both in text and figures. Line 84 also indicates that dBsm units are decibels referenced to a square meter. We also fixed a typo in Fig. 9 (prev fig 6). The maximum RCS indicated before in the label was of 28.28 dBsm, but it is actually 28.34 dBsm.

The introduced changes are even more confusing. The calibration terms characterize a ratio of a real measure over the calculated one. Therefore, calibration terms must be unitless in linear scale and in dB in the logarithmic scale. I do not understand what a calibration term in dB(AU^-1m^-2) means. The lines 72 – 76 are confusing. It is stated that kd is included to account for the units of the measured power which is in 10*log10(AU). One sentence later it is stated that kd is unitless. If it is unitless then the equation 1a has problems with units again. The nominator is unitless, the denominator has units of m^2*W as it was in the original version. I kindly ask the authors to carefully reconsider the units again. In fact, the previous version was better, the only problem was with units notation, i.e. dB was used instead
of dBm and dBsm (please see my previous comments). Please modify the units in such a way that the calibration factors are given in dB (unitless in linear scale). And please modify the units throughout the manuscript accordingly.

Following this suggestion, we now present the received power units to dBm and the calibration terms to dB. This led to the recalculation of the calibration terms absolute value.

Modifications:

- Because of this improvement, lines 79-80 and 82-84 explaining these arbitrary power units are no longer necessary and were removed.
- Eq. (2a) and line 542 were modified to remove the unnecessary digital gain term kd.
- Writing of the units corrected in lines 97, 112, 127, 142 and the Glossary.
- Absolute value of power/calibration terms corrected in lines 261, 264, 265, 267, 319-321, Table 1, and lines 512-531 of the calibration results.
- Additionally, Figures 3, 4(a), 5, 6 and 10 were modified to remain consistent with the power units.

**4.** This change is included in every mention of receiver losses as $L_r(T, r)$, and is therefore propagated to the RCS and reflectivity calibration terms as well, which now depend on temperature and range ($C_\Gamma(T, r)$, $C_Z(T, r)$).

I would recommend to use IF instead of r because for a different chirp configuration (slope) the relations between IF bins and range gates may change.

We originally used range because BASTA-Mini has only 4 standard operational modes, so it was not a very important distinction. Nevertheless, we agree that this formulation would improve the generality of the method, and consequently we modified all range dependent terms by terms depending on the IF frequency $F_b$.

Besides, we added a short explanation of the $F_b$ term in lines 75-81 and the new Equation (1), which indicates how $F_b$ it is associated with the range r.

**5. Section 5.5**

In this newly added section the authors, as far as I understand, assume that during the 'passive' observations the power variability along IF depends only on gain changes. In general case this is not true:

$Pr(IF) \sim G(IF)*(Tsys(IF) + Tamb)$

Here $Pr(IF)$ is the received power at IF in W, $G(IF)$ is the linear gain of the receiver chain at IF (unitless), $Tsys(IF)$ is system noise temperature at IF in K, Tamb is brightness temperature of the sky (or an object the radar was pointed to) in K, $\sim$ is the proportionality sign. From this equation one can see that the received power depends on two parameters, namely the gain and the system noise temperature. If I understand right, the authors did so-called single point calibration. Using the single point calibration is it not possible to separate the gain and the system noise temperature. Therefore, typically two-point calibrations are used in radars and radiometers. Also the authors need to know Tamb (at least with respect to $Tsys(IF)$). I kindly ask the authors to clarify how they took these aspects into account to calibrate all the IF bins of the radar receiver.

In this section we do not intend to estimate the absolute value of gain at the IF, but rather to quantify relative gain changes with respect to the calibrated IF frequency (associated with the target position). To do this using passive observations, we had to make the assumption of a constant noise power, both from the system and from environment in the 12 MHz bandwidth of the receiver. This assumption is reasonable because components used in the receiver have a much larger bandwidth (for example for the Low Noise Amplifier (LNA) it is of 35 GHz).

We indicated briefly this assumption in the previous version, but we didn't mention the impact it could have on uncertainty. This is now estimated from the LNA specifications. Its variability of gain and noise figure in the 12 MHz bandwidth used is smaller than 0.1 dB. Since LNA are typically the main source of system noise in the receiver, we consider that 0.1 dB is a safe estimation of the uncertainty introduced by assuming a constant system noise in the IF bandwidth. This term dominates the RMSE between the fit and data, and the inter-period variability, thus we now define the IF correction function uncertainty to be of 0.1 dB.

Despite this, we agree that two point calibrations are highly desirable because they enable the retrieval of receiver absolute gain and system noise. This is now indicated in the text.

Thus, article changes are:
- More accurate explanation of passive observations (lines 365-368, equation 9a)
- Brief indication of the benefits of performing two point calibrations for receivers (lines 370-373)
- Explanation of the constant system noise assumption and introduced uncertainty (lines 374-383, equation 9b)
- Clarification in the explanation on how the IF correction function is retrieved (lines 384-396, line 406-408).
- Final uncertainty of the IF correction function (lines 413-414)

**6.** However, we did another revision of the scanning data and concluded that, at present, it is not possible to retrieve alignment information with an accuracy comparable to the antenna beam-width. This is now stated in lines 294-295. The reason is that the repeatability of the scanner positioning is not sufficient to allow a reliable retrieval under our current procedure. Additionally, we now include a discussion on how parallax errors can influence the measurements (286-290), and indicate that calibration results are compatible with parallax errors smaller than the radar beamwidth (296-298). Since we don't have information on the exact alignment, we now mention the parallel antennas only as an hypothesis (245, 299-300). Finally, we improved the calibration methodology by indicating how parallax errors can be taken into account, suggesting the addition of an additional range dependent correction function (300-301), and by introducing an uncertainty term representing the error in the antennas alignment estimation (eq. 6b, lines 243-245 and 301-302).

The assumption on parallel antennas can lead to large uncertainties. The problem with two antennas is that it is possible to measure the pattern of the receiving antenna with an external transmitter but it is often not possible to measure the transmitting antenna. Basically, with the proposed method only two points of the possible range dependent bias are characterized.
Instead of leaving this large uncertainty source untouched, I would encourage the authors to make a relatively simple estimation of possible impacts (just theoretical calculations, taking into account different divergences (magnitude and direction) of the two antennas and bias measurements at two distances). This would definitely improve the quality of the manuscript. The result of this theoretical estimation would give a proxy for sigma_a in Eq. 6a which is currently, if I understand it right, completely neglected.

Just to better understanding I give a couple of figures:

[Figure]

On the left figure you can see different divergence directions. On the right figure I illustrate the impact (qualitatively). The authors could perform such calculations and give an estimate for sigma_a (maximum divergence from 0 dB line).

Thank you for this proposal, we believe it is a very good idea with good potential. Because of this, we performed several theoretical calculations to check if we could estimate a range of possible antenna misalignment angles and the associated uncertainty with our data. Summarizing, our results show that the experimental setup used is not appropriate for this measurement, but they also indicate us a path to perform such experiments in the future.

Since the targets used at 196 and 376.5 are different, the uncertainty in the **calibration coefficient difference** at both distances is very large (~3 dB). This uncertainty makes it impossible to bound the possible alignment within 1.5 degrees, which is our antenna characterization width. This large uncertainty comes mostly from the use of two different calibration targets. This decision was made because the experiment was designed to applicability of the absolute calibration method for different experimental setups, and because the proposed experiment was not considered at the time.

Given that the proposed experiment was not done during the campaigns presented in the article, we have no way to gather any additional information on antenna alignment for that period. Thus, we leave this section unchanged with respect to the previous version.

Yet, with the theoretical calculations we found that if we get an uncertainty in the order of 0.5 dB when comparing calibration constants at these two different distances, antenna misalignment could be constrained to values ranging in the order of tenths of degree. This could be achieved, for example, by using the 20 meter mast setup at both distances using the same reflector each time. It is worth noting that the tools developed for this analysis now enable the design of an experiment with optimized parameters for this retrieval.

Taking all this into consideration, in our opinion the potential of the proposed experiment indicate that it must be further studied for its implementation in future calibration campaigns.

**7.** To verify if data did follow a linear relationship, we did a new plot with the point density of all samples together. This figure has been added to the paper (Figure 7). In this figure it is easier to observe that deviated points are rather exceptional, with most points close to the regression.  From this figure we think the 0.13 dB RMSE value is representative for most samples. We also modified Figure 6 (D). Now it is only used to introduce the data set, with the linear fit shown in new Figure 7. This produced text changes in lines 351-355, and 360-371.

In Fig. 7 the authors just masked the problem I am talking about. I agree that a majority of samples follow the linear model. But some complete iterations (like green points in Fig 6d) are off by more than 0.5 dB.

This happens because, to capture the widest range of possible temperatures, we had to use longer time series of data. The use of longer time series introduced some points measured under suboptimal conditions. For example, we have observed that high wind speeds lead to larger variabilities in the calibration value due to oscillations of both the mast and the radar. Meanwhile, drizzle adds a time dependent bias, most likely caused by changes in wet radome (and wet target) attenuation over time. An effort was done to clean the dataset, but inevitably some noisy data points remained. This is now stated more clearly in lines 345 to 348.

Therefore, to estimate the temperature correction function and its uncertainty we did an statistical analysis of data, and then used the RMSE between the model and data points as the estimation of model uncertainty.

To bound the uncertainty value we calculated the RMSE of the model for each degree of deviation from the reference temperature, obtaining a RMSE range between 0.07 to 0.23 for 0 and +3 degrees of deviation respectively. We also checked the bias per degree for each iteration and for all the dataset, and found out that its mean value is always within +- 0.2 dB with respect to the model. Therefore, we now state that the temperature correction function uncertainty is less or equal to 0.23. This change is reflected in lines 353-357.

This is also true for the mentioned case, were most points are covered by the other iterations data. The larger spread reaching deviations of 0.5 dB in this case are caused by short period of drizzle that happened in this iteration. However, since this data, and the rest of data that deviates from the model is also included in the calculation of RMSE, we think this is a reliable criteria for the estimation of the temperature correction function uncertainty.

[revised manuscript text omitted]

---

## Author Response (AR3)

Response to the editor.

Dear Dr. Kneifel,

We carefully considered the comments of the reviewer and provide a point by point response below and indicate clearly changes made to the manuscript.

Regarding the concern about the correctness of Eq. 2., we would like to draw your attention to the fact that the formulation proposed by the reviewer does not correspond to the calibration definition we are using. The formulation proposed by the reviewer requires a theoretical estimation of the system power budget a priori, which is precisely what we want to avoid by using a reference reflector. The formulation that we use has been widely used previously in the literature.

We trust that the responses we provide below will convince you to accept our manuscript for publication. The discussions with the reviewer have been intense and we are grateful for the time and effort of the reviewer. His comments were very useful to improve the quality of our manuscript. However we would like to avoid longer delays in publishing our results.

Kind regards,

The authors

Response to the reviewer.

Dear Dr. Alexander Myagkov,

Once again we thank you for your attentive review, we think it helped us to achieve a complete work. Your last response was very detailed, enabling us to explain why we made certain decisions in our procedure.

Our previous responses are in blue
Your last comments are in green
Our new responses are in black

**1.- Prev. Author's response:** We agree about the need of characterizing the target to provide final uncertainty results. However, we do not agree that the lack of the target characterization cancels the validity of the results, since we used a theoretical model of target RCS that has all the properties that a calibrated target would have, except the absolute values. The results presented in the article enable the identification of several uncertainty sources, as well as
their relative contribution to the experiment uncertainty. This information can be very valuable to design future calibration experiments based on reference reflectors, whether they are mounted on masts or held by other means, such as UAVs. The underlying principles remain the same. We also add that our objective is not to claim we have a reference instrument, but to present all the information and advancements obtained from our experimental campaigns, specially in uncertainty characterization. For example, with the results we can quantitatively compare two different experimental setups, finding different factors limiting uncertainty for each (SCR for the 10 m mast, alignment for the 20 m mast). Additionally, we are not aware of any other published methodology of radar calibration that considers the bias introduced due to misalignment between target and radar. For us this work is a step towards more precise calibration methodologies, and we expect it to act as a reference to improve the preparation of future calibration experiments. Because it is true that at this stage we can only do a rough estimation of RCS uncertainty, we agree to highlight this explicitly.
This is now stated/included in:
Line 14 of the Abstract.
Lines 197-200
Line 229
Lines 507-509
Calibration result for all experiments (lines 512-531)
Table 2
Lines 585-587
This also implied modifications to some text in the article to remain consistent:
Lines 18-20 of the abstract.
Lines 54-55 of the Introduction
Lines 153-154

We also added an estimation of the maximum uncertainty in RCS characterization required to reach a calibration uncertainty of 0.5 dB in lines 592-597.

**Reviewer Comment:** Thanks for these changes. The modifications you have introduced make it much clearer for a reader that the main source of uncertainties is the characterization of the used reflectors and that this is not covered in this study. Just a few minor comments here:
- Please be consistent with the goal value. It is 1 dB in the abstract (line 20) but 0.5 dB throughout the text (e.g. lines 55, 593).

The goal value has been corrected, now is to reach uncertainty values under one dB, in the introduction (now line 53) and the conclusions (line 582).

- In lines 593 – 594 you apply the formula of the std of a sum of two uncorrelated variables, if I understand it right (sqrt(0.4^2+0.3^2) = 0.5). I would agree with it if one would use a newly manufactured reflector every time, but most likely, it will be just a single one so the bias due to the corner reflector will be constant (zero variance, the formula is not applicable). In this case the reflector contributes to the systematic error, while the effects considered in the manuscript characterize the random error. I would recommend making this clear.

What we mean by target uncertainty is the uncertainty in the target RCS characterization. When the target is characterized we get its absolute RCS value plus its uncertainty. The absolute RCS is input in the RCS calibration term calculation and uncertainty is propagated in the uncertainty budget (eq. (7a), term $\Gamma\sigma_0^2$).

The value of the required target uncertainty is now adjusted to be consistent with the goal of 1 dB in line 583.

Since the study makes conclusions on the total uncertainties with a number of assumptions (i.e. parallel antennas, flat noise figure, reflector with known cross section), please consider a change of the title of the manuscript to something like "Aspects of cloud radar calibration based on corner reflectors".

We did a modification to the title to make it more precise. Now the article title is "Absolute Calibration method for FMCW Cloud Radars based on corner reflectors". We think this title is appropriate because it reflects the content of the paper which presents a method to calibrate the absolute reflectivity of a FMCW cloud radar. Most uncertainty sources of this method have been quantified, and the remaining hypotheses are clearly indicated. We explain in the article the hypotheses of flat noise figure and known target cross section, and we indicate how they can be verified and corrected by using well known and established techniques. These corrections do not change the proposed method at all.

The only part that is not yet resolved is the assessment of the impact of non parallel antennas. We removed the mention of uncertainty assessment for antenna alignment errors in the abstract, and a new phrase is included in the conclusions to state explicitly that parallel antennas are a hypothesis (Lines 592-593).

**2.- Prev. Author's response:** Following this suggestion, we now present the received power units to dBm and the calibration terms to dB. This led to the recalculation of the calibration terms absolute value.
Modifications:
- Because of this improvement, lines 79-80 and 82-84 explaining these arbitrary power units are no longer necessary and were removed.
- Eq. (2a) and line 542 were modified to remove the unnecessary digital gain term kd.-Writing of the units corrected in lines 97, 112, 127, 142 and the Glossary.
- Absolute value of power/calibration terms corrected in lines 261, 264, 265, 267, 319-321, Table 1, and lines 512-531 of the calibration results.
- Additionally, Figures 3, 4(a), 5, 6 and 10 were modified to remain consistent with the power units.

**Reviewer Comment:** Here I still have a major concern. I would like to thank the authors for their efforts, but the equation 2 is still wrong. It cannot give dB units, because the ratio in the parenthesis is still not unitless. Please note, that this is already the third time I ask to adjust all the terms properly. I try to do my best to explain what I expect if I were using the method proposed by the authors. Typing long formulas in Word is a bit inconvenient. Therefore, I wrote my considerations on paper.

I forgot to write one term. Lat is the attenuation due to propagation to the target and back.

In my point of view, the authors should use the calibration term Cgamma as in Eq. 4 in the drawing. It is clear how to use it in order to correct the measurements (it works in the same way for power and reflectivity values) and it is unitless as it should be. I marked two components of the calibration term, the one characterized in the manuscript (greed rectangle) and the one with certain assumptions (red rectangle). I believe such a separation in the very beginning of the manuscript would help a reader to understand which effects are considered and which are not.

Thanks for clarifying your concern. Now we understand why it was difficult to agree on this point. What you are requesting does not correspond to the calibration definition we are using. We are calculating the weather radar calibration term (also commonly referred to as calibration constant, or radar system constant) by using the formulation of the following references:

- Bringi, V. N., and V. Chandrasekar. *Polarimetric Doppler weather radar: principles and applications*. Cambridge university press, 2001. Chapter 6, Section 6.3 "Radar Calibration".

- Skolnik, Merrill I. "Radar handbook third edition." *McGrawHill*, 2008. Chapter 19, Section 19.2 "The Radar Equation for Meteorological Targets".

- Chandrasekar, V., et al. "Calibration procedures for global precipitation-measurement ground-validation radars." *URSI Radio Science Bulletin* 2015.355, 2015: 45-73.

This term relates power measurements at a given distance with a physical quantity (RCS or reflectivity). The equation with their units are:
* * *
Linear scale:

$$RCS[m^2] \ = \ C_\Gamma[m^{-2} \cdot mW^{-1}] \cdot (r[m])^4 \cdot P_r[mW] \cdot (L_{at}(r))^2$$

$$Z[mm^6 \cdot m^{-3}] = C[mm^6 \cdot m^{-5} \cdot mW^{-1}] \cdot (r[m])^2 \cdot P_r(r)[mW] \cdot (L_{at}(r))^2$$
* * *
dB Scale:

$$RCS[dBsm] \ = \ C_\Gamma[dB(1 \cdot m^{-2} \cdot mW^{-1})] + 10 \cdot log((r[m])^4) + P_r(r)[dBm] + 2 \cdot L_{at}(r)[dB]$$

$$Z[dBZ] = C[dB(mm^6 \cdot m^{-5} \cdot mW^{-1})] + 10 \cdot log((r[m])^2) + P_r(r)[dBm] + 2 \cdot L_{at}(r)[dB]$$

Cg is the calibration term for RCS measurements, and C for equivalent reflectivity measurements (Z). r is the distance to the target in meters, Pr( r ) the received power from this target and Lat( r ) the atmospheric attenuation between target and radar (unitless).

We can see from these expressions that the calibration term we use cannot be unitless, because it has to transform distance and power measurements into square meters or millimeters to the sixth power per cubic meter for RCS and Z respectively. The correct SI units based on our references are 10 log(1 m$^{-2}$ mW$^{-1}$) for the RCS calibration and 10 log(1 mm$^6$ m$^{-5}$ mW$^{-1}$) for Z.

We also would like to clarify that the calibration constant you propose is not applicable to our method. What you propose is a correction term for an internal calibration. It requires a theoretical estimation of the system power budget a priori, which is precisely what we want to avoid by using a reference reflector. See for example:

Chandrasekar, V., et al. "Calibration procedures for global precipitation-measurement ground-validation radars." *URSI Radio Science Bulletin* 2015.355 (2015): 45-73. Section 3.1.

Yin, Jiapeng, et al. "UAV-aided weather radar calibration." *IEEE Transactions on Geoscience and Remote Sensing* 57.12 (2019): 10362-10375.

Thus, we now put the physical units of the calibration terms in lines 90-91, 105, 117. Also they are now included in the calibration results (lines 504 to 523) and in figures 6 (a) and 10.

We trust that our response should resolve the long discussion on this issue.

**3.- Prev. Author's response:** In this section we do not intend to estimate the absolute value of gain at the IF, but rather to quantify relative gain changes with respect to the calibrated IF

frequency (associated with the target position). To do this using passive observations, we had to make the assumption of a constant noise power, both from the system and from environment in the 12 MHz bandwidth of the receiver. This assumption is reasonable because components used in the receiver have a much larger bandwidth (for example for the Low Noise Amplifier (LNA) it is of 35 GHz). We indicated briefly this assumption in the previous version, but we didn't mention the impact it could have on uncertainty. This is now estimated from the LNA specifications. Ist variability of gain and noise figure in the 12 MHz bandwidth used is smaller than 0.1 dB. Since LNA are typically the main source of system noise in the receiver, we consider that 0.1 dB is a safe estimation of the uncertainty introduced by assuming a constant system noise in the IF bandwidth. This term dominates the RMSE between the fit and data, and the inter-period variability, thus we now define the IF correction function uncertainty to be of 0.1 dB. Despite this, we agree that two point calibrations are highly desirable because they enable the retrieval of receiver absolute gain and system noise. This is now indicated in the text. Thus, article changes are:
- More accurate explanation of passive observations (lines 365-368, equation 9a)
- Brief indication of the benefits of performing two point calibrations for receivers (lines 370-373)
- Explanation of the constant system noise assumption and introduced uncertainty (lines 374-383, equation 9b)
- Clarification in the explanation on how the IF correction function is retrieved (lines 384-396, line 406-408).
- Final uncertainty of the IF correction function (lines 413-414)

**Reviewer Comment:** A couple of major concerns:
- Lines 265-267: formulation is wrong. The very first component in the receiver chain is a low noise **amplifier**. The receiver amplifies the received signal not reduces the power level. Despite on the amplification LNA reduces the signal-to-noise ratio and this is characterized by its noise figure. I recommend to use the formula I gave last time for the proper explanation.

- Formula 9: If noise powers are in linear units, why dB values of losses are subtracted? If noise powers are in dBm they cannot be summed.

Thanks for noticing the issue of Eq. 9, it was incorrect. We modified it to remain consistent with the units used through the text, based on your proposed explanation. To remain consistent we also indicate that the receiver gain term is equivalent to the receiver loss term introduced in Section 2, Eq. 2a.

Thus, lines 350-353 and eqs. 9a and 9b are now corrected. Eq. (10) is rewritten to remain consistent in the use of linear units for the receiver loss.

Also a minor concern:
- Line 376: please be careful here. I completely agree that noise figure of LNA can be assumed flat. Since noise figure of mixers, active IF filters and ADC units can be very high, these components can still contribute to the noise figure despite the amplification of LNA. The question is how about other components and standing waves, which can produce wavy

shape (> 100 K variability) of the system noise temperature even within several MHz bandwidth? If this was considered please mention, if not please explain that other effects caused by other components of the receiving chain or standing waves can affect the assumption of the constant noise temperature.

- Taking into account the three comments, I recommend a revisiting the section 5.5 lines 364 384.

We did a calculation using the first three terms of the Friis formula to estimate the impact of variations in Noise Figure (NF) and gain after the LNA and the mixer, and in any realistic case the noise figure variability in the 12 MHz bandwidth is smaller than 0.1 dB.

For this estimation we used:
- The LNA gain at the operation frequency (>= 20 dB), and its corresponding noise figure of approx. 4 dB.
- The mixer conversion loss of 8.6 dB with a variability of 0.3 dB (most likely overestimated, since it corresponds to the mixer conversion loss variation between 95 and 96 GHz)
- An additional system noise figure representing the rest of the system, which was varied between 0 and 400 K (for the third term of the Friis formula).

For all cases, variations in the total NF remain smaller than 0.1 dB in the 12 MHz band (with a minimum of 4.07 to a maximum of 4.17 dB).

As proposed, our considerations are now stated explicitly. This introduced modifications in lines 359-378.

**4.- Prev. Author's response:** Thank you for this proposal, we believe it is a very good idea with good potential. Because of this, we performed several theoretical calculations to check if we could estimate a range of possible antenna misalignment angles and the associated uncertainty with our data. Summarizing, our results show that the experimental setup used is not appropriate for this measurement, but they also indicate us a path to perform such experiments in the future. Since the targets used at 196 and 376.5 are different, the uncertainty in the calibration coefficient difference at both distances is very
large (~3 dB). This uncertainty makes it impossible to bound the possible alignment within 1.5 degrees, which is our antenna characterization width. This large uncertainty comes mostly from the use of two different calibration targets. This decision was made because the experiment was designed to applicability of the absolute calibration method for different experimental setups, and because the proposed experiment was not considered at the time. Given that the proposed experiment was not done during the campaigns presented in the article, we have no way to gather any additional information on antenna alignment for that period. Thus, we leave this section unchanged with respect to the previous version. Yet, with the theoretical calculations we found that if we get an uncertainty in the order of 0.5 dB when comparing calibration constants at these two different distances, antenna misalignment could be constrained to values ranging in the order of tenths of degree. This could be

achieved, for example, by using the 20 meter mast setup at both distances using the same reflector
each time. It is worth noting that the tools developed for this analysis now enable the design of an experiment with optimized parameters for this retrieval. Taking all this into consideration, in our opinion the potential of the proposed experiment indicate that it must be further studied for its implementation in future calibration campaigns.

**Reviewer Comment:** I understand the point of the authors. I would recommend to do the following. Please summarize the effects which you have characterized in your study and write that the uncertainty you have found are only related to these effects. Also specify explicitly a list of effects to be characterized in the future (target, IF dependency of noise figure, parallelism of antennas, etc).

All of this is already stated in the abstract, in Section 4 and in the conclusions. However, we reformulated the sentence related to IF and target characterization in the conclusions for more clarity (see lines 594-597).

[revised manuscript text omitted]